# Timbral effects on consonance disentangle psychoacoustic mechanisms and suggest perceptual origins for musical scales

Raja Marjieh [1,2,6] ✉, Peter M. C. Harrison [2,3,6] ✉, Harin Lee[2,4], Fotini Deligiannaki[2,5] & Nori Jacoby [2] ✉

The phenomenon of musical consonance is an essential feature in diverse musical styles. The traditional belief, supported by centuries of Western music theory and psychological studies, is that consonance derives from simple (harmonic) frequency ratios between tones and is insensitive to timbre. Here we show through five large-scale behavioral studies, comprising 235,440 human judgments from US and South Korean populations, that harmonic consonance preferences can be reshaped by timbral manipulations, even as far as to induce preferences for inharmonic intervals. We show how such effects may suggest perceptual origins for diverse scale systems ranging from the gamelan's slendro scale to the tuning of Western meantone and equal-tempered scales. Through computational modeling we show that these timbral manipulations dissociate competing psychoacoustic mechanisms underlying consonance, and we derive an updated computational model combining liking of harmonicity, disliking of fast beats (roughness), and liking of slow beats. Altogether, this work showcases how large-scale behavioral experiments can inform classical questions in auditory perception.

Many musical styles involve multiple performers playing or singing simultaneously[1–3]. In Western music, this practice is underwritten by the notion of harmony, defining how multiple musical tones may be combined together into polyphonic sonorities or chords. To a given listener, certain chords will sound particularly pleasant, or consonant, while others will sound relatively unpleasant, or dissonant. This phenomenon has immense importance in many musical styles, determining how musical notes are organized into scales, how these scales are tuned, and how chords are constructed from these scales[4–8]. It has consequently drawn sustained attention from many researchers ranging from philosophers (Pythagoras) to mathematicians (Leibniz, Euler), music theorists (Zarlino, Rameau), and modern-day psychologists and ethnomusicologists[9–19].

Consonance perception is thought to derive from both psychoacoustic and cultural factors (e.g. ref. 20). Several psychoacoustic mechanisms have been proposed over the years, including fusion[21,22] and combination tones[23–25], but the two main extant theories attribute consonance either to interference between partials[26] or to harmonicity detection[19] (see refs. 12,20,27 for a similar conclusion). Both theories predict that chords comprising harmonic tones should sound most pleasant when the tones are related by harmonic pitch intervals (i.e., those that correspond to simple frequency ratios, e.g., 2:1, 3:2). This would explain why many (though not all) scale systems across the world seem to have developed to favor harmonic pitch intervals[5,28]. Within a given society, cultural familiarity with particular musical styles will further contribute to consonance perception, biasing listeners

[1]Department of Psychology, Princeton University, Princeton, NJ, USA. [2]Max Planck Institute for Empirical Aesthetics, Frankfurt am Main, Germany. [3]Centre for Music and Science, University of Cambridge, Cambridge, UK. [4]Max Planck Institute for Human Cognitive and Brain Sciences, Leipzig, Germany. [5]German Aerospace Center (DLR), Institute for AI Safety and Security, Bonn, Germany. [6]These authors contributed equally: Raja Marjieh, Peter M. C. Harrison. ✉e-mail: raja.marjieh@princeton.edu; pmch2@cam.ac.uk; nori.jacoby@ae.mpg.de

toward preferring sonorities that occur often within a given musical style. Styles based on Western tonality will therefore reinforce preferences for harmonic pitch intervals[20,29], but styles with different harmonic systems may induce different biases[13,30].

To this date, a crucial unsolved question has been whether consonance perception depends on the timbre of the underlying chord tones. Previous literature provides contradictory perspectives here. Traditional Western music theory implies that consonance should be independent of timbre; it provides just one scheme for categorizing intervals into consonances and dissonances, and this scheme applies equally to all musical instruments[18]. On the other hand, prominent psychoacoustic theories of consonance (in particular, Helmholtz's interference theory) imply that consonance judgments should vary substantially depending on the positions and magnitudes of the tones' upper harmonics[26]; an argument which was also reiterated in later papers[31,32]. Nevertheless, both Helmholtz[26] and Sethares[31] provide only theoretical arguments, and while Plomp and Levelt[32] cite an old empirical paper claiming to show one such effect[33], the latter paper failed to replicate with modern methods[34]. Furthermore, recent decades of psychological studies seem to show that timbral manipulations do not qualitatively affect consonance judgments[12,27,34–36]; though see[37] for a study of the effect of timbre on the statistical learning of melodic grammars.

Here we address this question with a series of 23 large-scale behavioral experiments comprising 4272 online participants and 235,440 human judgments (participants were allowed to participate in multiple experiments, but only once within a given experiment; see Supplementary Tables 1 and 2 for a breakdown of the number of unique participants in each experiment). These experiments have three important features:

First: continuous treatment of pitch intervals. Previous consonance research has used stimuli drawn solely from discrete scales, most commonly the Western 12-tone chromatic scale[12,27,29,34,35,38,39]. This is problematic because it neglects potentially interesting structure in between the scale degrees of the chromatic scale, and because the resulting paradigm is inherently Western-centric. Here we instead avoid making any assumptions about scale systems, and instead take advantage of novel psychological techniques (dense rating; Gibbs Sampling with People[40]) to construct continuous consonance maps directly from behavioral data.

Second: systematic exploration of timbral features. Several recent consonance studies have included timbral manipulations, but generally only explored a limited number of manipulations[12,34,35] or used manipulations designed to demonstrate generalizability rather than to test particular hypotheses[27]. Here we take a more systematic approach. We focus in particular on spectral manipulations, because (as we show later) these yield particularly clear hypothetical effects in computational modeling. In a series of studies, we address the three main ways of manipulating a harmonic spectrum: (1) changing the frequencies of the harmonics (Studies 2 and 5), (2) changing the amplitudes of the harmonics (Study 3), and (3) deleting individual harmonics entirely (Study 4). In Study 1 we establish an experimental baseline for harmonic dyads (two-note chords), in Studies 2–4 we explore timbral features in the context of dyads, and in Study 5 we generalize our results to triads (three-note chords).

Third: concurrent computational modeling. Previous research has developed many computational models operationalizing different theories of consonance perception. Here we use such models to understand what predictions different theories should make for different spectral manipulations. We focus in particular on the two psychoacoustic models that performed best in a recent systematic evaluation of almost all extant consonance models[20]. The first is the model of Hutchinson and Knopoff[41], which supposes that dissonance derives from unpleasant interactions between neighboring partials in the frequency spectrum, potentially corresponding to the fast beats

that occur when two tones of similar frequencies are superposed. Specifically, the model assigns a dissonance (or roughness) score for each pair of partials based on a parametric function that depends on the frequency difference between them, and then combines those scores additively. The second is the harmonicity model of Harrison and Pearce[42], which supposes that chords become consonant when they align well with an idealized harmonic series. The harmonicity score is calculated by computing similarity scores between different idealized harmonic templates and a compact representation of the chord spectrum. We confirm the robustness of the modeling results by running supplementary analyses with a collection of alternative interference models[31,43] and harmonicity models[44,45] (Supplementary Fig. 1). Additionally, we plot results from a new composite model that comprises a simple weighted average of updated versions of the interference and harmonicity models, with weights fixed throughout the paper, and show it can account for results that are not explained by any of the models in isolation.

Together, these 23 experiments characterize the relationship between timbre and consonance in great detail, shedding light on the psychological mechanisms underpinning consonance perception, as well as the close connection between musical instruments and the cultural evolution of musical styles.

## Results

### Baseline results for harmonic dyads (Study 1)

Following a long tradition of music theory and music psychology research, we begin by studying the consonance of two-tone chords (dyads) as a function of the frequency ratio between those tones (e.g. refs. 26,27,38,41,46). We represent those frequency ratios as pitch intervals, where the pitch interval in semitones is calculated as $12 \log_2 \frac{f_2}{f_1}$, where $f_1$ and $f_2$ are the two frequencies. In a subsequent study (Study 5) we then generalize the approach to three-tone chords (triads). Throughout these studies we investigate the moderating role of timbre.

Study 1A characterizes dyadic consonance perception for synthetic harmonic complex tones. These tones are constructed by combining pure tones (i.e., simple sinusoids) whose frequencies are all integer multiples (i.e., harmonics) of a common fundamental frequency. Such tones have long been used as idealized approximations of the pitched sounds produced by the human voice and by common musical instruments (e.g. refs. 32,41,47).

In each trial of the experiment, we played US participants ($N = 198$) dyads comprising two harmonic complex tones (10 harmonics per tone, 3 dB/octave spectral roll-off, 1.3 s in duration), sampling the pitch interval between the two tones from a uniform distribution over the range of 0–15 semitones, and sampling the pitch of the lower tone from a uniform distribution over the range G3-F4. The participant was then asked to rate the dyad's pleasantness on a scale from 1 (completely disagree) to 7 (completely agree) (Fig. 1a). Following previous research, we use pleasantness as a convenient synonym for consonance that is understood well by nonmusicians (e.g. refs. 13,27,48). Other possible synonyms exist (e.g., smoothness, purity, harmoniousness, tension, stability), but in practice human ratings tend to be correlated across these synonyms[48]. We collected many ratings from many participants for many dyads, and summarized the results using a Gaussian kernel smoother (bandwidth: 0.2 semitones), finding consonance peaks using a peak-picking algorithm and constructing 95% confidence intervals using nonparametric bootstrapping (see "Methods").

Figure 1b summarizes the results from Study 1A (see Supplementary Movie 1 for a video version). While the trial-level data (light blue points) are relatively noisy, highlighting the subjectivity of the perceptual evaluations, the averaging process reveals clear structure that aligns with the Western discrete scale system, as well as traditional hierarchies of consonance and dissonance. In particular, we find eight clear peaks in the pleasantness judgments; these peaks are located

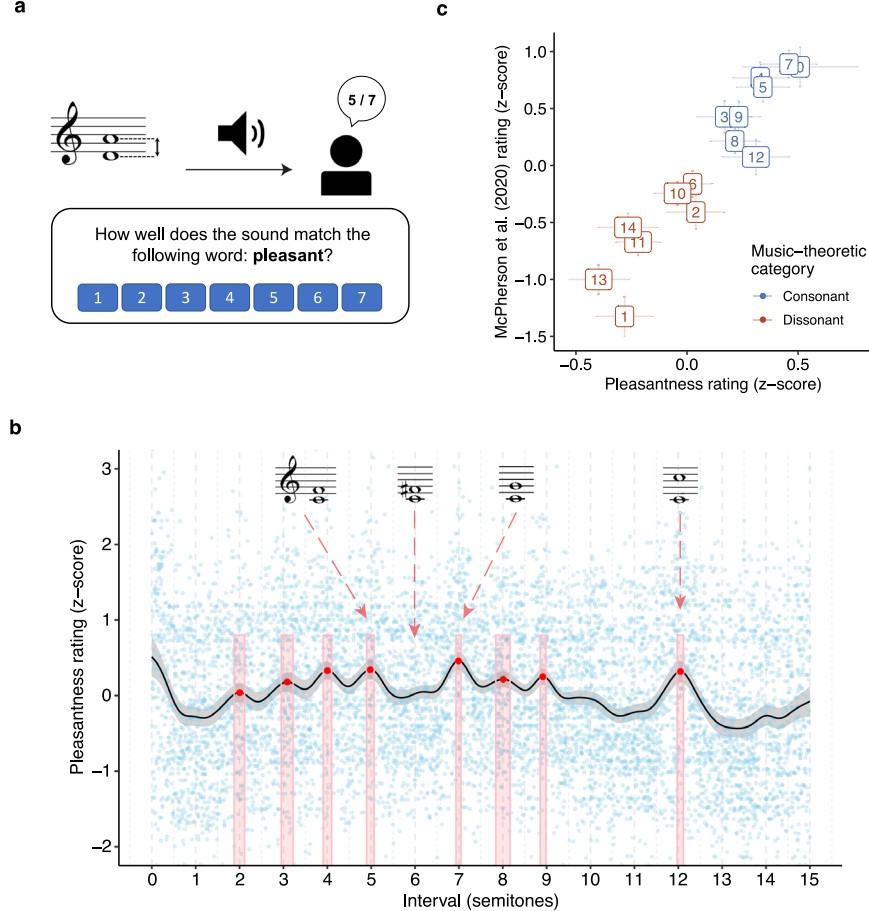

**Fig. 1 | Dyadic consonance for harmonic complex tones (Study 1A, *N* = 198 participants). a** Schematic illustration of the rating task. **b** Consonance profile (black line) derived through kernel smoothing (*z*-scored, Gaussian kernel, bandwidth 0.2 semitones, 95% confidence interval), superposed on raw data (blue points). Peaks estimated by a peak-picking algorithm are marked in red as mean values ±95% confidence intervals (bootstrapped, 1000 replicates). **c** Comparing smoothed ratings at integer intervals to traditional music-theoretic classifications and to data from McPherson et al.[39] (*N* = 100 participants) (*z*-scored over participants, data presented as mean values ±95% bootstrapped confidence intervals).

close to the integer semitones that make up the Western 12-tone chromatic scale (average distance of 0.05 semitones, 95% confidence intervals = [0.03, 0.08] semitones; random chance would give 0.25 semitones). The relative heights of these peaks replicate traditional music-theoretic classifications of intervals into consonant (blue) and dissonant (red) categories (Fig. 1c; mean difference between consonant/dissonant intervals is 0.38 standard deviations, 95% confidence intervals = [0.29, 0.46]). The results also correlate very well with the results of previous behavioral experiments studying the relative consonance of different Western intervals, including aggregated results from seven laboratory studies from the late nineteenth/early twentieth centuries[46] (*r*(10) = 0.96, *p* < 0.001, 95% confidence intervals = [0.85, 0.99]), a recent laboratory study by Bowling et al.[38] (*r*(10) = 0.91, *p* < 0.001, 95% confidence intervals = [0.71, 0.98]), and a recent online study by McPherson et al.[39] (*r*(13) = 0.94, *p* < 0.001, 95% confidence intervals = [0.84, 0.98]). Lastly, a Monte Carlo split-half correlation analysis showed an excellent internal reliability (*r* = 0.87, 95% confidence intervals = [0.74, 0.94], 1000 permutations). Together, these results give us confidence in the reliability and validity of our experimental methods.

Synthetic harmonic complex tones are traditionally intended as approximations to the kinds of complex tones produced by real musical instruments. To verify that they elicit similar consonance profiles, we conducted three follow-up experiments repeating Study 1A but with tones from three synthetic musical instruments: flute, guitar, and piano (Study 1B, 602 participants). We find that these

instruments indeed produce broadly comparable consonance profiles to the idealized harmonic complex tones (Fig. 2; mean *r* = 0.56, mean *ρ* = 0.62) though with certain peaks falling either side of the 95% statistical significance cutoff for different instruments (e.g., the minor 7th peak, 10 semitones, is only statistically significant for the guitar).

The differences we see between these instruments are potentially interesting but difficult to interpret definitively, because the instruments vary on multiple factors, including both temporal and spectral features. In the rest of this paper, we therefore focus on artificial harmonic complex tones, manipulated in very precise and interpretable ways, so that we can better distinguish the underlying causal factors that affect consonance perception.

Previous research has documented how consonance judgments can vary as a function of musical experience (e.g. refs. 32,49). In all of our experiments, we therefore asked our participants to report their years of musical experience, allowing us to estimate the sensitivity of our results to musical expertise. Figure 3 plots results for Study 1A differentiated by musical expertise (median split: 2.0 years of musical experience). We found, in general, that participants with different levels of musical experience gave qualitatively similar results. Participants with more musical experience (>2 years, musicians) tended to give more differentiated judgments than participants with less musical experience (≤2 years, nonmusicians) (mean *SD* of *z*-scored musician profiles = 0.23, 95% confidence intervals = [0.18, 0.28], mean SD of *z*-scored nonmusician profiles = 0.15, 95% confidence

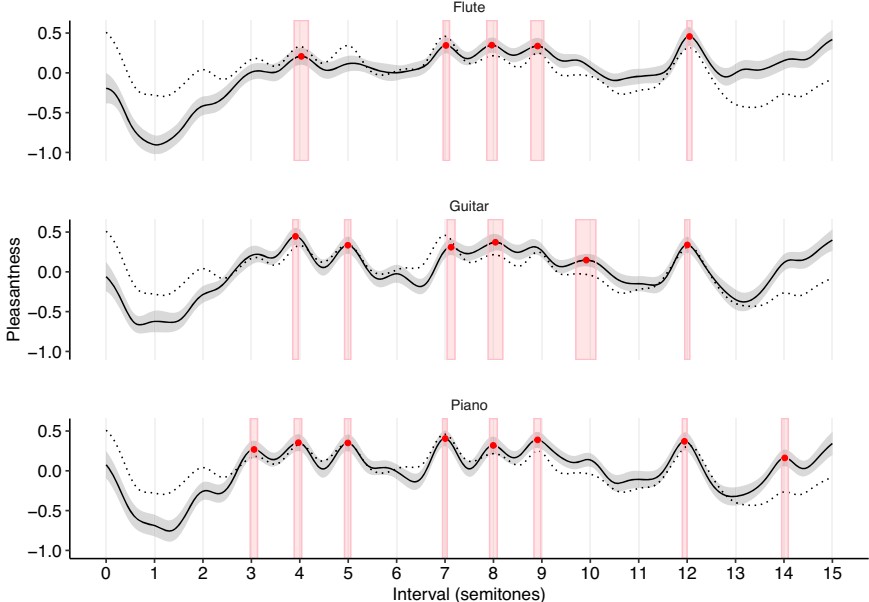

**Fig. 2 | Dyadic consonance for synthesized Western instruments (Study 1B; flute: *N* = 190 participants, guitar: *N* = 210 participants, piano: *N* = 198 participants).** Consonance profiles for the Western instruments (*z*-scored) are plotted as mean values ± 95% bootstrapped confidence intervals (bandwidth of 0.2 semitones, 1000 bootstrap replicates). The consonance profile for harmonic complex tones (Study 1A) is plotted as a reference dotted line.

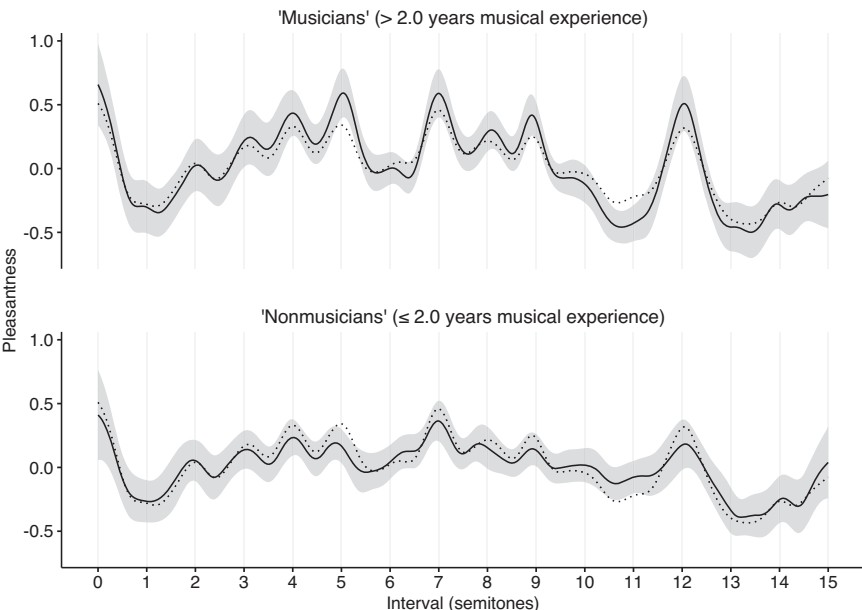

**Fig. 3 | Dyadic consonance for harmonic complex tones as a function of musicianship (Study 1A, *N* = 198 participants).** Consonance profiles (*z*-scored) are plotted as mean values ± 95% bootstrapped confidence intervals (bandwidth of 0.2 semitones, 1000 bootstrap replicates). The musicianship threshold (2.0 years) corresponds to a median split of the participant group. The reference dotted line corresponds to the consonance profile derived from the full participant group.

intervals = [0.11, 0.18], mean difference = 0.08, 95% confidence intervals = [0.07, 0.10], bootstrapped over experiments), but in general the judgments correlated quite highly across both groups (mean $\rho$ of 0.68, 95% confidence intervals = [0.57, 0.80], bootstrapped over experiments). This consistency may be due to universal psychoacoustic processes, but it may also be due to the sophisticated implicit musical knowledge that listeners are known to develop even in the absence of formal musical training[50,51]. In the following studies, we focus on analyzing data aggregated over all musical experience levels.

## Changing harmonic frequencies (Study 2)

We begin by considering how consonance judgments may be altered if we change the frequencies of the harmonics that make up the complex tone. Such effects have been previously hypothesized in previous work[15] but not yet empirically tested.

We first consider a stretching manipulation proposed by Sethares[15], where we manipulate the spacing between the harmonics in the complex tone (Study 2A). Similar kinds of stretching are present to small degrees in string instruments as a consequence of string stiffness, causing slight inharmonicity[52]. We define the frequency of the *i*th

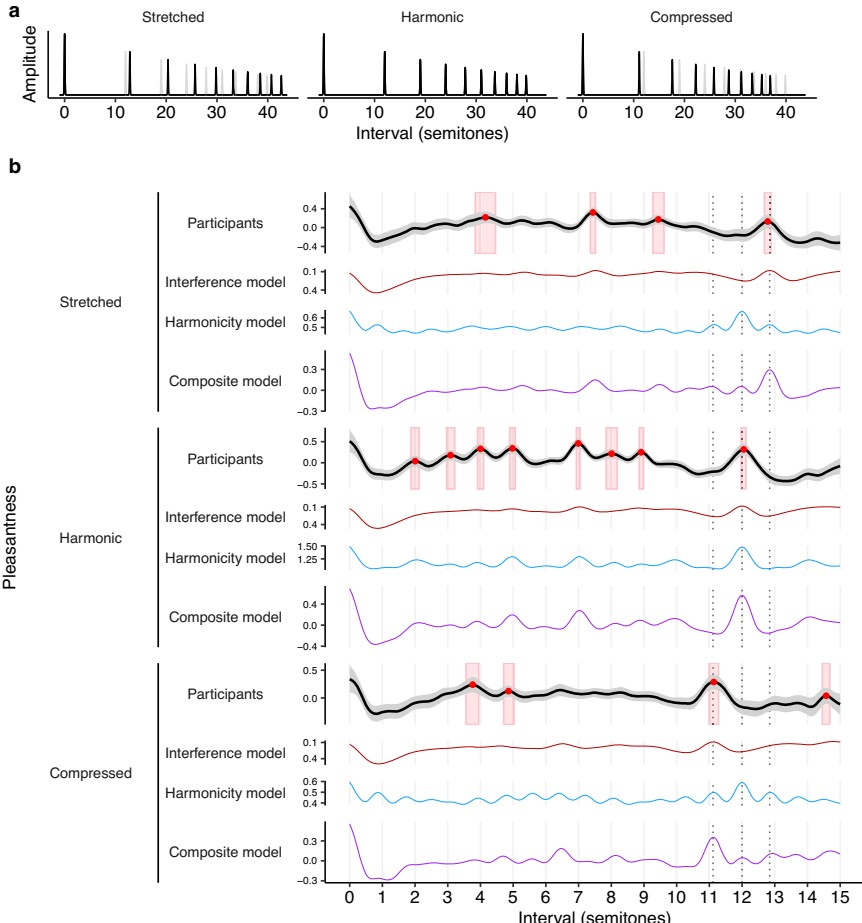

**Fig. 4 | Spectral stretching/compression and consonance (Studies 1A, 2A).**
**a** Stretched and compressed tone spectra, with a baseline harmonic spectrum
(gray) for comparison. **b** Dyadic pleasantness judgments for stretched ($N = 194$
participants), harmonic ($N = 198$ participants), and compressed ($N = 202$ partici-
pants) tones. Behavioral results are summarized using a kernel smoother with a
bandwidth of 0.2 semitones, with 95% confidence intervals (bootstrapped, 1000
replicates) shaded in gray, peak locations plotted as red circles with red rectangles
indicating mean values ± 95% confidence intervals. Dotted lines indicate the loca-
tion of the compressed, harmonic, and stretched octaves.

partial as $f_i = f_0 \gamma^{\log_2(i+1)}$, where $f_0$ is the fundamental frequency and $\gamma$
is the stretching parameter: $\gamma = 1.9$ then defines a compressed tone,
$\gamma = 2$ defines a standard harmonic tone, and $\gamma = 2.1$ defines a stretched
tone (Fig. 4a).

Interference models predict that this spectral stretching/com-
pression manipulation should yield analogous stretching/compression
in consonance profiles (e.g., Fig. 4b, red lines; see Supplementary
Figs. 2 and 3 for equivalent results from alternative models). Intuitively,
this can be understood from the observation that interference is
minimized when partials from different tones align neatly with each
other; if we then stretch each tone's spectrum, we must also stretch the
intervals between the tones to maintain this alignment.

Interestingly, harmonicity models do not predict such an effect;
instead, they generally predict that these manipulations will largely
eliminate pleasantness variation, and any residual variation will still be
located at harmonic intervals (e.g., Fig. 4b, blue lines; see also Sup-
plementary Figs. 2 and 3). Once the individual tones become inhar-
monic, the overall chord also becomes inharmonic, irrespective of the
intervals between the tones.

We conducted a pair of experiments to construct consonance
profiles for stretched (Study 2A(i), 194 participants) and compressed
(Study 2A(ii), 202 participants) tones, and compared these to baseline
profiles for harmonic tones (Study 1A, 198 participants). As predicted
by the interference account, we find that we can indeed induce pre-
ferences for stretched and compressed intervals, in line with the

corresponding spectral manipulations (Fig. 4b; see Supplementary
Movies 2 and 3 for video versions). For example, for dyads comprising
stretched tones, we clearly see preferences for stretched octaves (peak
at 12.78, 95% confidence intervals = [12.68, 12.88] semitones; an
unstretched octave would be 12.00 semitones) contrasting with the
results from harmonic tones (peak at 12.04, 95% confidence inter-
vals = [11.97, 12.11] semitones) (Fig. 4b). We see similar stretching/
compression for other consonant intervals, though in some cases the
peaks lose clarity for the inharmonic tones. These effects are con-
sistent with the predictions of the interference models, but not with
the predictions of the harmonicity models; the results therefore pro-
vide evidence that interference between partials is an important con-
tributor to consonance perception, in contrast to recent claims in the
literature that interference is irrelevant to consonance
perception[13,27,38,53]. However, the results are not inconsistent with the
idea that harmonicity makes some contribution to consonance per-
ception; we see in particular that the composite consonance model
successfully predicts all of the stretching/compression phenomena
(Fig. 4b; note that whenever we plot the composite model we plot it in
its final form, including modifications motivated by later experiments).

We were interested in testing whether these effects are specific to
US listeners. We therefore ran a replication experiment of the
stretching/compressing dyadic consonance study (Study 2A) as well as
the harmonic baseline (Study 1) with Korean participants ($N = 68$)
recruited with the aid of a research assistant in the local area (Study

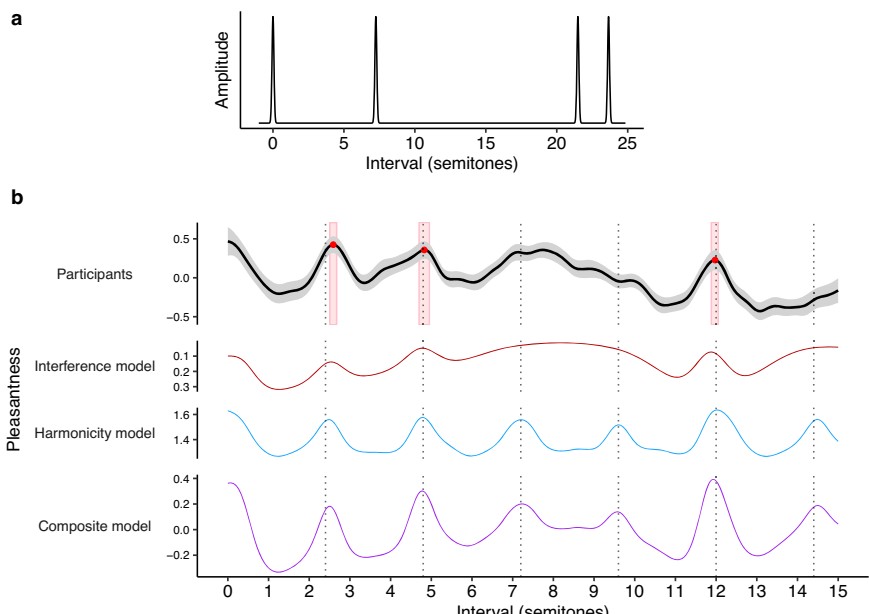

**Fig. 5 | Dyadic pleasantness judgments for the bonang (Study 2C, *N* = 170 participants). a** Idealized spectrum for the bonang (Sethares[15]). **b** Pleasantness judgments (95% confidence intervals) for dyads comprising a harmonic complex tone (lower) combined with an idealized bonang tone (upper) (Study 2C). Peak locations are plotted as red circles with red rectangles indicating mean values ± 95% confidence intervals (bootstrapped, 1000 replicates). Interference[41] and harmonicity[42] model predictions are plotted for reference. The slendro scale, approximated as 5-tone equal temperament[15], is plotted with dashed lines; note how this scale barely overlaps with the 12-tone scale.

2B). Participants were required to be native Korean speakers and resident in South Korea; all experiment instructions were translated to Korean by a native speaker. The resulting consonance profiles are shown in Supplementary Figs. 4–6 and Supplementary Movies 4–6. The general effect clearly replicates with the new participant group: stretched spectra produce a stretched consonance profile, whereas compressed spectra produce a compressed consonance profile.

Other kinds of inharmonicity can be produced by certain percussion instruments, for example the metallophones of Indonesian gamelans and the xylophone-like *renats* used in Thai classical music[15]. Each instrument will have an idiosyncratic spectrum that reflects its particular physical construction, with potentially interesting implications for consonance perception.

In Study 2C we investigate an inharmonic tone inspired by one such instrument, the *bonang*. The bonang is an instrument from the Javanese gamelan comprising a collection of small gongs. In order to achieve arbitrary microtonal pitches, we use a synthetic approximation to the bonang proposed by Sethares[15] on the basis of field measurements, corresponding to four equally weighted harmonics with frequencies of $f_0, 1.52f_0, 3.46f_0$, and $3.92f_0$ (Fig. 5a). Following Sethares[15], we play dyads where the upper tone corresponds to this idealized bonang, and the lower tone corresponds to a standard harmonic tone with four equally weighted harmonics. This combination is intended to reflect a common kind of texture in Javanese gamelan music, where the inharmonic bonang is played alongside a harmonic instrument or voice.

The results from the corresponding dyad rating experiment are plotted in Fig. 5b (Study 2C, 170 participants; see Supplementary Movie 7 for a video version). As with the harmonic tones, we see a clear pleasantness peak at the octave, 11.98 semitones, 95% confidence intervals = [11.88, 12.05]. However, the other peaks previously seen at harmonic intervals are now either missing or displaced to inharmonic locations. In particular, we see clear peaks at 2.60, 95% confidence intervals = [2.51, 2.67] semitones and 4.80, 95% confidence intervals = [4.70, 4.95] semitones, neither of which are harmonic intervals. Conversely, we do not see any peaks at the major third (no peak detected in

95% of bootstrap samples within the interval [3.5, 4.5]) or the perfect fifth (no peak detected in 76% of bootstrap samples within the interval [6.5, 7.5]). These peaks are each predicted by the interference model and the harmonicity model; however, the harmonicity model also predicts several additional peaks that do not manifest clearly in the behavioral data (see also Supplementary Fig. 7 for additional models).

Sethares[15] made an interesting claim that the two main scales of Javanese gamelan music (the *slendro* scale and the *pelog* scale) reflect the consonance profiles of its instruments (see also refs. 54,55). In particular, he proposed that the inharmonic *slendro* scale might be explained in terms of the consonance profile produced by combining a harmonic complex tone with a bonang tone, as in our own experiment, emulating the interaction between the human voice and the bonang. We have correspondingly annotated Fig. 5b with the locations of the slendro scale degrees, approximating the scale as 5-tone equal temperament[15,56]. As predicted by Sethares[15], we do find that the empirical consonance curve aligns neatly with the slendro scale, even though our Western participants are likely to have no or little exposure to Javanese gamelan music; in particular, each observed peak (2.6, 4.8, and 12.0 semitones) is located close to a slendro scale degree (2.4, 4.8, and 12.0 semitones). Interestingly, while the two remaining slendro scale degrees (7.2 and 9.6 semitones) do not have corresponding behavioral peaks, they do have corresponding peaks in the harmonicity curve.

To summarize, we found that manipulating the frequencies of the harmonics can induce inharmonic consonance profiles. In particular, stretching/compressing the harmonic series leads to stretched/compressed consonance profiles (Study 2A–B), whereas replacing the upper dyad tone with a synthetic bonang tone yields an idiosyncratic consonance profile that aligns with the slendro scale from the Javanese gamelan (Study 2C), even for participants with little or no prior experience with this scale. The stretching/compressing manipulation is interesting from a modeling perspective, because it clearly dissociates the predictions of the interference and the harmonicity models, and shows that only the former are compatible with the behavioral data. The latter manipulation is interesting from a cultural evolution perspective, because it supports the hypothesis that the

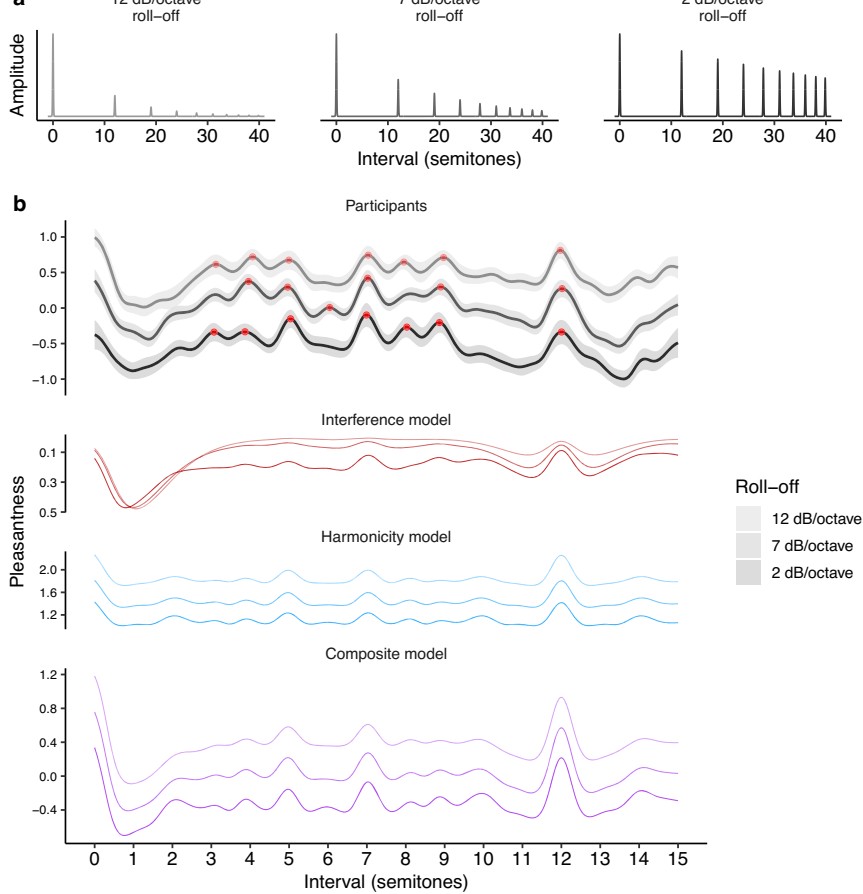

**Fig. 6 | Dyadic consonance as a function of roll-off (Study 3, N = 322 partici-pants). a** Tone spectra for three representative levels of roll-off (12, 7, and 2 dB/octave). **b** Pleasantness judgments (mean values ± 95% confidence intervals bootstrapped with 1000 replicates) for these three roll-off levels computed using kernel smoothing (bandwidth = 0.2 semitones), plotted alongside interference[41] and harmonicity[42] model predictions. Peak locations are plotted as red circles.

slendro scale developed in part as a specific consequence of the acoustic properties of Javanese gamelan instruments[15,54,55].

### Changing harmonic amplitudes (Study 3)
We now consider how consonance profiles may be affected by changing the *amplitudes* of their tones' harmonics. In particular, we focus on the so-called *spectral roll-off* parameter, which determines the rate at which harmonic amplitude rolls off (decreases) as harmonic number increases. For example, a tone with high roll-off might have amplitude decrease at a rate of 12 dB per octave, whereas a tone with low roll-off might have amplitude decrease at only 3 dB per octave (Fig. 6a).

Interference theories would predict that roll-off has a major influence on consonance judgments: increased roll-off should reduce the magnitude of beating effects induced by the upper harmonics, hence producing flatter consonance profiles (Fig. 6b, red lines). In contrast, harmonicity theories predict that pleasantness profiles should remain highly differentiated. Harmonicity comes primarily from integer relationships between fundamental frequencies, a phenomenon which is relatively robust in the face of roll-off manipulations (Fig. 6b, blue lines).

Interference and harmonicity theories also predict potential main effects of spectral roll-off on pleasantness (with higher overall pleasantness rating for larger roll-off parameters). According to the interference models, almost every interval elicits some interactions between upper harmonics, and hence becomes more pleasant with increased roll off (Fig. 6b, red lines; standardized regression coefficients ($\beta$) for the main effects = 0.39, 0.83, 0.62 for the three models).

Harmonicity models make less consistent predictions: some predict an overall main effect on pleasantness (e.g., Fig. 6b, blue lines; $\beta = 0.91$), whereas others do not predict a strong effect ($\beta = 0.02, 0.23$; see also Supplementary Figs. 8–10).

We tested these predictions in Study 3 with a dense dyad rating experiment (322 participants) manipulating both pitch interval (0–15 semitones) and roll-off (0–15 dB roll-off/octave, Fig. 6a; see Supplementary Movies 8–10 for video versions). We see a clear main effect of roll off (as predicted by both theories), with participants finding greater roll-offs more pleasant (Fig. 6b). However, we see no clear effect on pleasantness variability; the profiles remain highly differentiated for all roll-off levels. Indeed, we find that a generalized additive model using just main effects of roll-off and of pitch interval can explain 98% of the variance of smoothed consonance ratings, indicating that, despite its strong main effect ($\beta = 0.89$), roll off has a minimal effect on the shape of the consonance profile.

These results are clearly inconsistent with the interference model, which predicted that the consonance profile should lose its differentiation at higher levels of spectral roll-off. In contrast, the results are highly consistent with the harmonicity model, which predicted that the profile's differentiation should remain preserved.

The results are also evidently inconsistent with any composite model that averages the interference and harmonicity models, because that model will also end up being overly sensitive to roll-off manipulations. How might we solve this problem? One solution is to increase the interference model's sensitivity to low-amplitude harmonics. The original model[41] takes interference as being proportional

to amplitude squared (i.e., intensity); if we reduce the exponent (e.g., from 2.0 to 1.3), the interference model becomes less sensitive to roll-off manipulations. We incorporate this update in the composite model that we plot throughout this paper, and we see that the model predicts the data in the present experiment very well (Fig. 6).

## Deleting entire harmonics (Study 4)

Study 3 found that reducing harmonic amplitudes had little effect on the shape of consonance profiles, in contrast to the predictions of the interference model. In Studies 4A and 4B, we now ask whether consonance profiles are affected by a more radical manipulation: completely deleting particular harmonics.

Some previous studies have investigated the effects of harmonic deletion, with mostly negative results. Vos[57] assessed subjective purity judgments in the neighborhood of the major third and the perfect fifth, and found that purity ratings increased when removing the even harmonics from the upper tone, but the overall shape of the consonance profiles remained broadly similar. Nordmark and Fahlén[35] took the minor ninth dyad, and investigated the effect of deleting the partials in each tone theoretically responsible for the most interference; however, they found no effect on consonance judgments. McLachlan et al.[34] tried removing (1) all even harmonics and (2) all harmonics above the fundamental frequency for a collection of dyads, but likewise found no clear effect on consonance judgments.

On the face of it, these historic results seem to be conclusive evidence against interference theories of consonance, which clearly predict that the consonance of non-unison intervals should depend on the existence of upper harmonics[34,35]. However, there are a couple of reasons to be skeptical here. One is that these historic studies do not test intervals in between the scale degrees of the 12-tone chromatic scale, and therefore potentially miss certain changes to the consonance profiles. The second is that historic studies using pure tones (i.e., tones with no upper harmonics) have often identified different (typically flatter) profiles to those that used complex tones (e.g. ref. 32).

We therefore reexamined this question in Study 4A (485 participants). We focused on three tone types (Fig. 7a): a tone with five equally weighted harmonics (bottom row), a tone with the third harmonic deleted (middle row), and a pure tone with all upper harmonics deleted (top row). We chose this combination of tones because (1) it is possible to delete the third harmonic without changing the tone's spectral centroid (i.e., mean spectral frequency), which is useful because the spectral centroid is known to contribute to pleasantness[10], and because (2) pure tones are good for making comparisons with prior studies (e.g. refs. 27,34).

We see a clear effect of the manipulation: deleting harmonics reduces the number of peaks in the pleasantness profiles, producing smoother and less differentiated curves (Fig. 7b; see Supplementary Movies 11–13 for video versions). In particular, the peak-picking

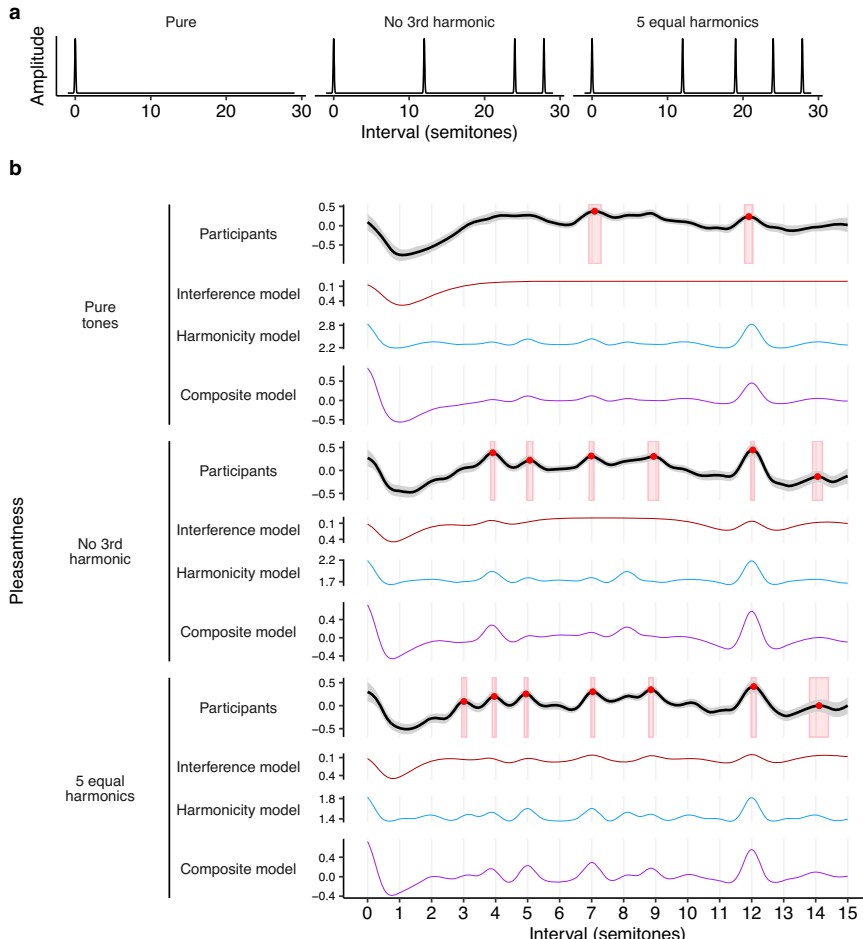

**Fig. 7 | Harmonic deletion and consonance (Study 4A). a** Tone spectra used in Study 4A. **b** Pleasantness judgments (95% confidence intervals bootstrapped with 1000 replicates) for dyads using these tone spectra (pure: $N = 176$ participants; no 3rd harmonic: $N = 160$ participants; 5 equal harmonics: $N = 149$ participants), plotted alongside interference[41] and harmonicity[42] model predictions (Study 4A). Peak locations are plotted as red circles with red rectangles indicating mean values ± 95% confidence intervals.

algorithm identifies seven statistically reliable peaks for the full spectrum (minor third, major third, perfect fourth, perfect fifth, major sixth, octave, and major tenth); deleting the third harmonic causes the minor third peak to be lost, and deleting the remaining upper harmonics causes all but the perfect fifth and octave peaks to be lost.

At a first glance, this picture is consistent with the predictions of both interference and harmonicity models: both predict that deleting harmonics will remove peaks from the pleasantness profiles. However, closer inspection yields interesting discrepancies. For example, the interference model erroneously predicts a complete absence of peaks for pure tones, and so a harmonicity component is necessary to predict the behavioral peak at the perfect fifth (7 semitones) and the octave (12 semitones). Furthermore, the harmonicity model correctly predicts that deleting the third harmonic should eliminate the pleasantness peak at the minor 3rd (3 semitones), unlike the interference model. Both models successfully predict that deleting the remaining harmonics should eliminate the behavioral peak at the major 3rd (4 semitones), but this also eliminates the behavioral peak at the perfect 4th (5 semitones), something which is not predicted by either model. In summary, the results point to an important contribution of harmonicity to consonance perception, but there are certain details of the timbral effects that still fail to be explained by existing models. Moreover, these discrepancies are not obviously solved by averaging the models to form the composite model.

As a follow-up question, we wondered what effect this timbral manipulation would have on preferred tunings for particular musical intervals. If listeners do indeed prefer different tunings for different musical timbres, this would imply that there is no such thing as an ideal tuning system for maximizing consonance, and instead the ideal tuning system should depend on the timbres being used.

What tunings should Western listeners ordinarily consider most consonant? On the one hand, interference and harmonicity models predict that consonance should be maximized by *just-intoned* intervals,

which correspond to exact simple integer ratios (e.g., 2:1 for the octave, 3:2 for the fifth, and so on). On the other hand, Western listeners typically have substantial experience with 12-tone equal temperament, the most common tuning system in Western music. Therefore, we might expect them to consider *equal-tempered* intervals most consonant.

How should harmonic deletion affect these tuning preferences? Interference theories clearly predict that listeners' preferences for just-intoned intervals should be eliminated by this manipulation, on account of eliminating the beating between upper harmonics (Fig. 8, red lines). Harmonicity theories meanwhile predict that consonance preferences peaks at *just-intoned* intervals should still be possible in the absence of upper harmonics; however, in practice certain harmonicity models do predict that consonance preferences will be somewhat flattened by deleting upper harmonics (Fig. 8, blue lines).

Previous work studying chordal tuning preferences has been limited to discrete tuning comparisons (e.g., Pythagorean tuning versus equal-temperament) and often to very small participant groups (e.g., 10 participants)[58,59]. Our work instead uses much larger participant groups and systematically evaluates a continuous range of pitch intervals (Study 4B, 1341 participants). In order to maximize statistical power for our timbral comparisons, we focus on just two tone types: (1) harmonic complex tones with low spectral roll-off (3 dB/octave), and (2) pure tones (corresponding to infinite spectral roll-off). We collect ratings in the neighborhood of several prototypical consonances from Western music theory: the major third (5:4), the major sixth (5:3), and the octave (2:1). We chose the major third and the major sixth because they have significantly different tunings in just intonation versus equal temperament, which is helpful for distinguishing the candidate theories. We additionally chose the octave because of previous work indicating listener preferences for stretched octaves (e.g. refs. 60–62).

The results for harmonic tones were considerably more interesting than we anticipated (Fig. 8a, solid lines; see Supplementary Movies 14–16 for video versions). For both the major sixth and the

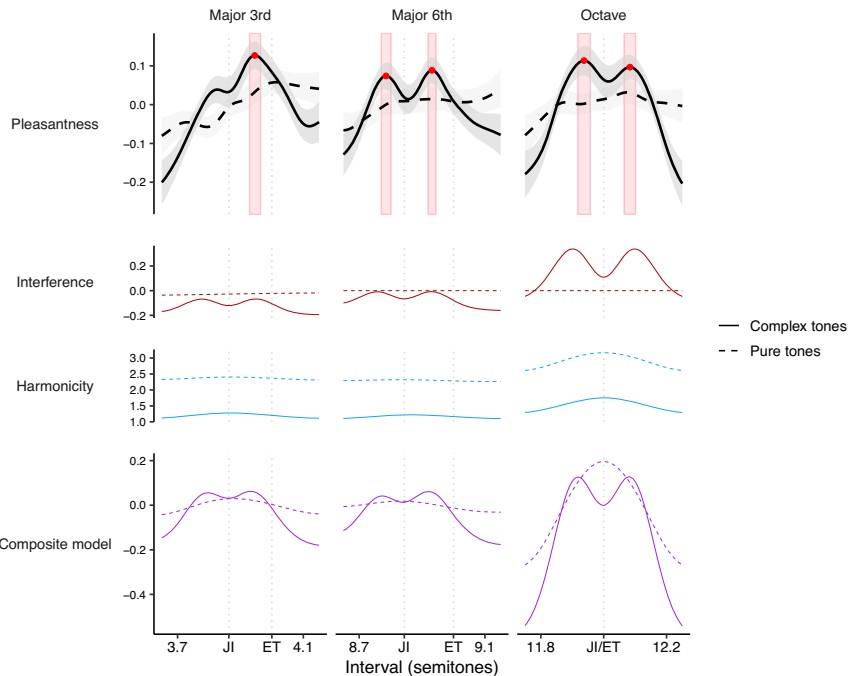

**Fig. 8 | Dyadic preferences for (mis)tunings of the major 3rd (harmonic: *N* = 237 participants; pure: *N* = 266 participants), major 6th (harmonic: *N* = 230 participants; pure: *N* = 227 participants), and the octave (harmonic: *N* = 196 participants; pure: *N* = 185 participants) (Study 4B).** Pleasantness judgments (*z*-scored within participants, 95% confidence intervals bootstrapped with 1000 replicates) are plotted alongside interference[41] and harmonicity[42] model predictions. Note that the between-groups design means that judgments for complex tones and pure tones are *z*-scored independently (but always within participants). Peak locations are plotted as red circles, with red rectangles indicating mean values ± 95% confidence intervals for these peak locations (bootstrapped, 1000 replicates). Just-intoned and equal-tempered versions of each interval are marked with JI and ET respectively.

octave, we see two clear peaks that sit either side of the just-intoned interval (8.78, 95% confidence intervals = [8.77, 8.80] and 8.93, 95% confidence intervals = [8.92, 8.94] for the major sixth, and 11.94, 95% confidence intervals = [11.92, 11.96] and 12.08, 95% confidence intervals = [12.07, 12.1] for the octave). In neither case do these peaks overlap with the equal-tempered interval. For the major third, we see a less clear version: the peak on the sharp side of just intonation is clear (3.95, 95% confidence intervals = [3.93, 3.96]), but the equivalent peak on the flat side is more subtle, only being detected in 66% of the bootstrap iterations. Nonetheless, the pattern of results is clearly inconsistent with simple preferences for just intonation or equal temperament.

Why might listeners prefer slight deviations from just intonation? One explanation would be that listeners positively enjoy the slow pulsating beats that these slight deviations induce for the feeling of richness that they convey, as hypothesized by Hall[6]. However, this concept is missed in the modeling literature; current interference models assume that beating is universally disliked[31,41,43].

If these preference patterns are indeed explained by enjoyment of slow beats, then they should be eliminated by deleting the upper harmonics from the complex tones to produce pure tones. Indeed, this is what we see (Fig. 8, dashed lines): the preference for slight deviations from just intonation is eliminated, resulting in relatively flat consonance curves (see Supplementary Movies 17–19 for video versions).

This preference for slow beats can be incorporated into the Hutchinson-Knopoff model by modifying its dissonance kernel such that small critical bandwidths yield negative dissonance (i.e., pleasantness; Supplementary Fig. 11). Incorporating this modification enables the composite consonance model to reproduce the observed preference for slight deviations from just intonation (Fig. 8).

Overall, the results are consistent with a composite model as long as it incorporates the adjustments to the Hutchinson-Knopoff interference model described above. The main limitation of the current implementation is that it underpredicts the extent to which removing harmonics flattens the octave peak. However, the model correctly captures the peak flattening for the other harmonic intervals (i.e., major 3rd and major 6th).

To summarize: Studies 4A and 4B demonstrate that deleting upper harmonics substantively influences consonance judgments. The pattern of effects in Study 4A (intervals ranging from 0 to 15 semitones) is predicted fairly well by the harmonicity model but not by the interference model. The pattern of effects in Study 4B (tuning preferences) is meanwhile not directly explained either by existing interference or harmonicity models; however, they do seem to be explainable by resorting to a slow beats theory that can easily be incorporated into interference models.

### Generalizing to triads (Study 5)

The previous studies definitively show that spectral manipulations can affect consonance perception for two-note chords (dyads). However, much of Western music is built from chords comprising at least three notes (triads, tetrads, etc.). It is worth asking whether these spectral effects can generalize to these larger chords.

The dense rating techniques used in the previous studies work well for the one-dimensional domain of dyadic intervals, but they scale less well to higher-dimensional chords such as triads. As the number of dimensions increases, the behavioral ratings are spread out over increasingly wider spaces, making the local averages at any one point less and less reliable.

Here we therefore use an alternative method called GSP[40]. This method coordinates participants into collaboratively exploring the stimulus space to find regions of (in this case) high consonance. In our application the stimulus space is two-dimensional, and corresponds to a space of possible triads. Each trial begins at a point on this plane, with the participant being presented with a slider corresponding to either

horizontal or vertical motion in the plane. Moving and releasing the slider triggers a new chord to be played corresponding to the updated position in the plane. The participant is told to move the slider to maximize the chord's pleasantness (Fig. 9a); when they are finished, the chord is then passed to next participant, who then manipulates the other interval and passes the chord to the next participant, and so on for a chosen number of iterations (typically 40) (Fig. 9b). Trials from many participant chains starting at many different points are averaged using a kernel density estimator.

Figure 9c shows baseline GSP results for harmonic triads with 3 dB/octave spectral roll-off (Study 5A, 228 US participants). Analogous to the prior results for dyads (Fig. 1), the present results now reflect the Western consonance hierarchy for triads. Some of the structure is inherited directly from the dyadic profile; for example, we see a clear diagonal line corresponding to chords whose intervals sum to 12 semitones (e.g., [5,7],[7,5] where the two numbers correspond to the intervals between the lower two notes and the upper two notes respectively). We also see hotspots corresponding to prototypical triadic sonorities, especially the three inversions of the major triad ([4,3],[3,5],[5,4]). A further consonance hotspot corresponds to the bare fifth, or power chord, a chord comprising the fifth but no third. This chord is common in both medieval music and rock music. Looking specifically at locations corresponding to the Western 12-tone scale, we further find that our results correlate well with the most comprehensive available reference dataset[38] ($r$(52) = 0.73, $p < 0.001$, 95% confidence intervals = [0.57, 0.83]). Additionally, a Monte Carlo split-half correlation analysis showed an excellent internal reliability ($r$ = 0.93, 95% confidence intervals = [0.91, 0.96], 1000 permutations).

Figure 10 shows GSP results for stretched and compressed tones (Study 5B, 462 participants). Similar to the dyad results (Study 2A), we see that these tones elicit stretched and compressed consonance profiles respectively. This effect is visually prominent in the case of the octave diagonal, a line running from the middle top to the middle right of the consonance plot corresponding to chords whose lower and upper tones are separated by an octave. For harmonic tones, this diagonal is located at 12.04, 95% confidence intervals = [11.87, 12.21] semitones; for stretched tones, the diagonal shifts to 12.80, 95% confidence intervals = [12.76, 12.83] semitones, whereas for compressed tones it shifts to 11.20, 95% confidence intervals = [11.15, 11.25] semitones. As before, these results are clearly predicted by the interference model but not by the harmonicity model. In summary, Study 5 clearly replicates the results of Study 2A, but generalizes them from dyads to triads.

## Discussion

In this paper we sought to understand how consonance perception depends on the spectral properties of the underlying chord tones. We used dense rating and GSP to measure consonance judgments for continuous intervallic spaces (Study 1), systematically varying the spectra of the underlying chord tones (Studies 2–5), and interpreting the results using computational models of interference and harmonicity[41,42]. Our results show that these spectral manipulations do indeed influence consonance perception, with implications for our understanding of its underlying mechanisms.

Studies 2A–C investigated the effect of changing harmonic frequencies. We found that such manipulations can induce inharmonic consonance profiles, contrasting with the vast majority of previous consonance research, which consistently demonstrates either preferences for harmonic intervals[27,29,38,46] or (for some populations) an absence of any preferences[13]. In particular, Studies 2A and 2B showed that stretching or compressing tone spectra induced stretched and compressed consonance profiles respectively; this finding is particularly interesting because it is predicted by current interference but not harmonicity models of consonance. Study 2C subsequently showed that tone spectra modeled after the Javanese bonang also yield an inharmonic consonance profile; interestingly, as hypothesized by

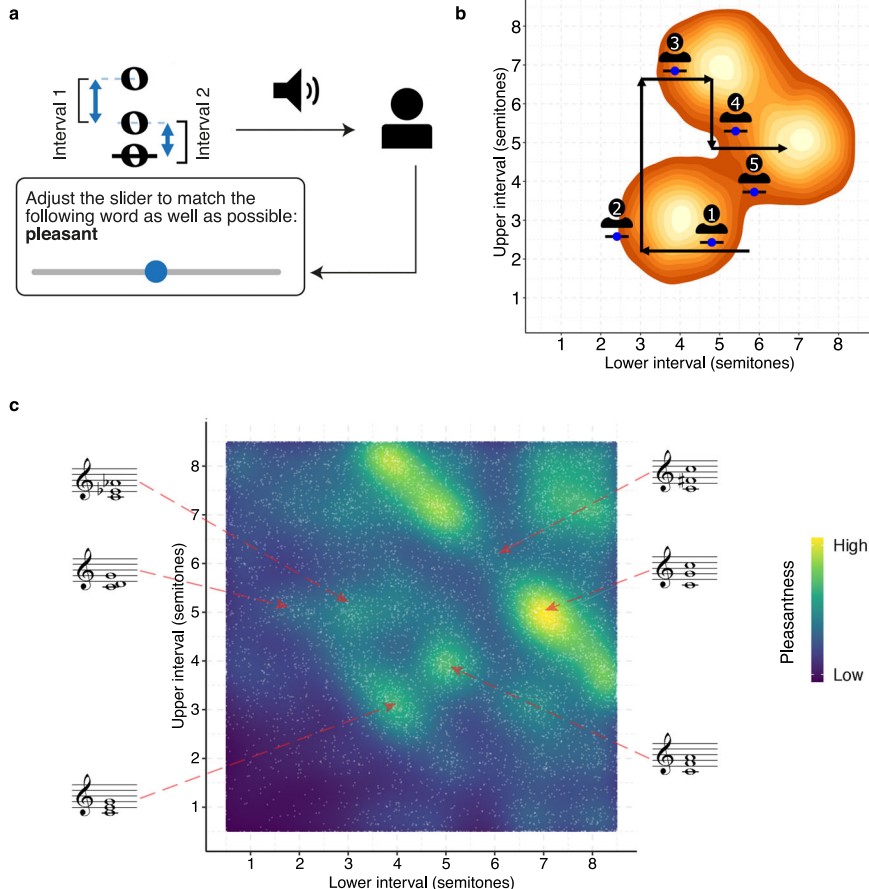

**Fig. 9 | Gibbs Sampling with People (GSP). a** Schematic illustration of the GSP task. **b** Example trajectory from a GSP chain overlaid on a kernel density estimate. **c** GSP data for harmonic triads (Study 5A, $N = 228$ participants). Individual iterations are plotted as white dots, whereas the KDE (bandwidth = 0.375) is plotted as a heat map, with light areas corresponding to high density/consonance.

Sethares[15], this consonance profile aligns with an idealized slendro scale as used in the Javanese gamelan. These results provide an empirical foundation for the idea that cultural variation in scale systems might in part be driven by the spectral properties of the musical instruments used by these different cultures[15,54,55]. They also provide an empirical justification for certain practices in the experimental music tradition of Dynamic Tonality[63,64], where tone spectra and scale tunings are manipulated in tandem (see also ref. 65).

Study 3 investigated the effect of changing harmonic amplitudes. In particular, we studied a *spectral roll-off* manipulation, changing the rate at which amplitude decays as a function of harmonic number. Interference models predict that increasing roll-off should gradually flatten the consonance profile, such that all intervals between four and ten semitones become similarly consonant. In contrast, harmonicity models predict that the consonance profile should be relatively robust to this manipulation. In actuality, the empirical data displayed no such flattening, only a general preference for higher roll-off, consistent with the harmonicity model.

Studies 4A and 4B investigated the effect of deleting harmonics entirely. First, Study 4A studied effects of harmonic deletion in the context of intervals spanning 0–15 semitones. We found clear effects of these manipulation, corresponding to the disappearance of certain peaks from the consonance profile. The precise pattern of peak elimination seemed best predicted by the harmonicity rather than the interference models. Second, Study 4B studied harmonic deletion in the context of fine-grained preferences for different tunings of particular consonant intervals. We identified a subtle but interesting phenomenon for tones with strong upper harmonics, whereby listeners

prefer slight deviations from exact just intonation, as hypothesized by Hall[6]. This effect seems explainable by listeners enjoying slow beats, a phenomenon that can be straightforwardly incorporated into interference models of consonance perception. Consistent with this hypothesis, these preferences for slight deviations disappeared upon elimination of the upper harmonics, presumably because this eliminates the slow beating effect.

Study 5 then investigated whether spectral manipulations could also affect consonance profiles for chords comprising more notes. In particular, we tested the stretching/compressing manipulation from Study 2A, and applied it to triads using the adaptive technique GSP[40]. We successfully replicated the stretching/compressing effect from Study 2A, showing that stretched/compressed tones yielded preferences for stretched/compressed chords respectively. Importantly, the triad space is much more complex than the dyad space, so it serves as a hard test for the composite model's predictions.

These results are provocative in the context of past centuries of Western music theory and empirical psychology studies, which hold that consonance derives from harmonic frequency ratios and is largely independent of timbre (e.g. refs. 34,35). The relationships we document between timbre and consonance are particularly interesting for explaining the cultural evolution of inharmonic scales in non-Western musical traditions, for example linking the slendro scale to the inharmonic percussion instruments of the Javanese gamelan, as suggested by Sethares[15]. The preferences we document for slight inharmonicity moreover have implications for the historical development of Western tuning systems: they indicate that the slight impurities of systems such as mean-tone and equal temperament may not always detract from

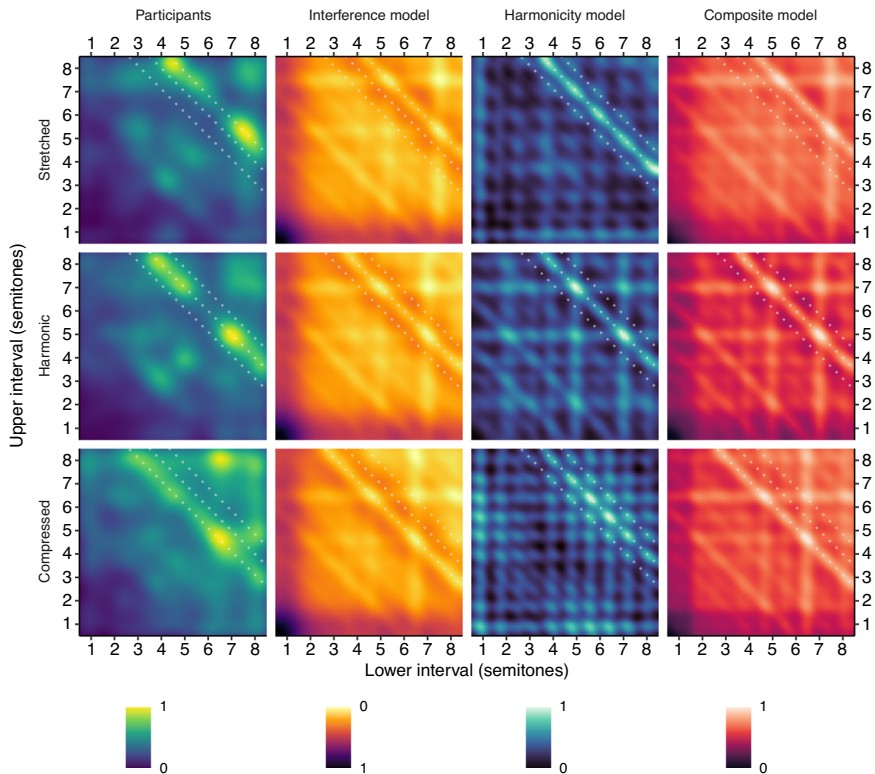

**Fig. 10 | Triadic pleasantness judgments for stretched and compressed tones (Study 5B; stretched: N = 229 participants, compressed: N = 233 participants).** GSP results are summarized using a KDE with a bandwidth of 0.375 semitones, and plotted alongside interference[41] and harmonicity[42] model predictions. Analogous pleasantness judgments for harmonic tones (Study 5A; harmonic: N = 228 participants) are also included as a reference. The theoretical locations of the compressed, harmonic, and stretched octave diagonals are plotted as dotted white lines. Values within each panel are scaled to the unit interval [0, 1].

subjective pleasantness, as is typically assumed, but can positively contribute to it by creating pleasant slow beating. For example, the perfect fifth C4-G4 elicits no perceptible beats when played in just intonation, but elicits pleasant slow (2.43 Hz) beats between the 2nd and 3rd harmonics if played in quarter-comma meantone tuning, and at 0.89 Hz if played in 12-tone equal temperament. In contrast, the major third elicits no perceptible beats in either just intonation or quarter-comma meantone tuning, but elicits fast (10.38 Hz) beats between the 4th and 5th harmonics in 12-tone equal temperament, a speed at which the beating will likely start to feel unpleasant again.

The combined results also have implications for competing theories of consonance perception. Unitary explanations—for example, that consonance is completely due to interference between partials[32], or completely due to harmonicity perception[27]—seem untenable in the context of these results. The stretching/compressing results from Studies 2A and 5B cannot be explained by current harmonicity models, only by interference models; the preferences for slight deviations from just intonation in Study 4B seem best explained by interference effects; conversely, the effects of deleting individual harmonics in Study 3 are best explained by harmonicity modeling, and the robustness to spectral roll-off demonstrated in Study 4 is consistent with an underlying harmonicity mechanism. Overall, it would seem plausible that consonance perception in Western listeners derives from a combination of (negatively valenced) interference and (positively valenced) harmonicity, as manifested in our composite model of consonance perception, which provides a generally good account of all the experiments described here.

These experiments also suggested several improvements to the underlying computational models. The results of Study 3 (spectral roll-off) motivated a modification to the interference model, making dissonance proportional to sound amplitude rather than sound intensity

(see also Vassilakis[43] for a discussion of theoretical issues concerning the connection between amplitude and dissonance). The results of Study 4B (tuning preferences) motivated a second modification, incorporating a liking of slow beats into the interference model. This liking of slow beats contrasts with the dislike of fast beats that is thought to drive the dissonance of more inharmonic intervals.

While Western listeners might have certain tendencies toward liking or disliking certain acoustic attributes (e.g., harmonicity), it is important to recognize that such appraisals are not necessarily universal. Musical enculturation seems to be important here, as evidenced for example by perceptual studies with the Tsimane' people, who seem (unlike Western listeners) to be indifferent toward harmonicity[13]. Acoustic attributes such as roughness and slow beating should therefore be interpreted as complementary perceptual ingredients that different musical styles can deploy to different ends (e.g. ref. 66).

While most of our experiments focused on Western listeners, we did perform a cross-cultural replication of our stretching/compressing manipulation with Korean listeners (Study 2B). This comparison is limited in that most of our participants will still have had some exposure to Western music, and moreover Korean music is just one of many musical cultures across the world. Nonetheless, these results provide initial evidence that timbral effects on consonance do generalize cross-culturally. There is a clear gap for more systematic cross-cultural research here; our methods should generalize well to such applications, especially given previous cross-cultural studies successfully using both rating scales[13] and slider interfaces[67].

Consonance perception is known to vary between individuals, even when the individuals are drawn from a relatively homogeneous cultural group[27,49]. It would have been interesting to analyze our own data at the participant level, but in practice our participant-level profiles had prohibitively low reliability due to the large stimulus space

and the relatively small number of trials per participant (the median split-half reliability in Study 1 was 0.39). As a compromise, in Fig. 3 we break down the results by musicianship. We see that the results were generally similar across the participant groups, though a systematic investigation might well yield more interesting conclusions. An interesting future path is to collect more data per participant and then use unsupervised clustering methods to identify population subgroups with different response strategies (e.g. ref. 68).

Our experiments primarily used artificial tones constructed through additive synthesis, giving us fine control over individual parts of the frequency spectrum, and facilitating the application of consonance models that take idealized frequency spectra as their input. However, we are excited to generalize our modeling and psychological experiments to more naturalistic sounds; we are currently developing experiments using audio samples from real pipe organs and pitched bells, both of which should elicit interesting consonance profiles according to our modeling.

One interesting manipulation that we did not study here is dichotic versus diotic presentation. Several papers have argued that presenting dyads dichotically (i.e., one tone to the left ear, and the other tone to the right ear) eliminates interference effects (e.g., roughness), because the two tones no longer interact in the auditory periphery[27,69]; comparing dichotic and diotic consonance profiles should therefore isolate the contribution of interference to consonance. Our dense rating and GSP methods should generalize well to this dichotic/diotic manipulation; however, we did not run such experiments here because it is difficult to ensure perfect dichotic stimulus presentation in online experiments.

We studied the consonance of isolated chords, but in many musical styles consonance is treated as a dynamic phenomenon; for example a dissonant note may be prepared by a previous chord and resolved by subsequent melodic movement. These temporal aspects of consonance are very interesting in their own right and worthy of further research (e.g. refs. 29,70). Nonetheless, we have shown how experiments with isolated chords are very effective for elucidating the fundamental relationships between intervallic structure and consonance. These relationships have shaped the treatment of musical pitch in a vast number of musical styles across the world, guiding the evolution of scales, harmonic vocabulary, and harmonic progressions[4–8].

We focused on a particular term of evaluation: pleasantness. This term has often been used in previous research as a way of capturing consonance perception in listeners without musical training (e.g. refs. 13,27,53), but there are many related terms that we could equally probe that are near-synonyms for consonance and deliver highly correlated evaluations, for example harmoniousness, attractiveness, purity, fit, and so on (e.g. refs. 48,71). Perhaps more interestingly, one could also probe a broader range of subjective connotations that capture wider aspects of tonal perception, for example emotional connotations such as happiness (e.g. ref. 72) and nostalgia (e.g. ref. 73).

While we have focused on identifying acoustic features that drive pleasantness in musical contexts, it is worth considering how these features might occur in other auditory contexts, and what the implications might be for the subjective appraisal of non-musical sounds. Harmonicity is a characteristic feature of vocalizations, and it has been suggested that social animals should find such sounds attractive because they (often) indicate the presence of conspecifics[9,38]. Roughness is meanwhile a feature of vocalizations indicating distress, in particular screams; it is evident that such sounds should cause discomfort because they are indicative of nearby danger (e.g. ref. 74). It is less clear why slow beats might sound attractive, but one speculation is that these fluctuations are suggestive of relaxed (i.e., non-distressed) vocalizations, and hence imply a safe environment. Alternatively, slow beating might be preferred simply because it makes the sounds more interesting. These hypotheses would imply that consonance is not simply a musical phenomenon, but instead is the (culturally shaped)

musical manifestation of deep principles from general auditory perception. However, these preferences are likely also to be shaped by musical experience; for example, familiarity with Western music may well encourage disliking of roughness and liking of slow beats.

Psychological research into consonance and other psychoacoustic domains has traditionally prioritized in-person data collection, where participants come to the laboratory and take behavioral experiments under supervision. In contrast, our work used online data collection, where participants took part remotely over the Internet. This involves surrendering some control over listening conditions which can only be partly recouped through headphone pre-screening tests and the like. Nonetheless, we found in practice that we could achieve very reliable group-level data this way, as indexed both by high internal reliability metrics (correlations ranging from 0.87 to 0.93) and high correlations with previous laboratory experiments (correlations ranging from 0.73 to 0.96). Furthermore, the online modality allowed us to scale up data collection to orders of magnitude greater than previous studies, enabling us to construct these detailed consonance maps that probe continuous interval spaces, as well as systematically exploring many timbral manipulations to prove our different consonance models. We believe that this approach has exciting potential to contribute to other classical research questions in domains such as psychology, psychophysics, and auditory perception.

## Methods

The present research has been conducted in compliance with approved Max Planck Society Ethics Council protocols (2020_05 and 2021_42).

### Paradigms

We use two behavioral paradigms in this paper: *dense rating* and *GSP*. The special feature of these paradigms is that they do not make any a priori assumptions about culturally specific scale systems, but instead characterize consonance as a smooth function over continuous space.

**Dense rating**. In the dense rating paradigm, participants are played chords that are randomly and uniformly sampled from continuous intervallic space. For *dyads* (chords comprising two tones), we typically study intervals in the range [0, 15] semitones; in successive trials we might therefore see dyads such as 4.87 semitones, 12.32 semitones, or 1.83 semitones. Each trial receives a pleasantness rating on a scale from 1 (completely disagree) to 7 (completely agree) (Fig. 1a). We then summarize the results from all the trials using a Gaussian kernel smoother. The degree of smoothing is set by the bandwidth parameter, which can be set by the experimenter in order to achieve an arbitrary balance between bias and variance: decreasing the bandwidth allows the smoother to capture more detail (less bias) at the cost of lower reliability (higher variance). To help with the interpretability of the data, we use a single bandwidth parameter for all experiments. In particular, we use a bandwidth of 0.2 semitones for all experiments spanning 0–15 semitones, and a bandwidth of 0.035 semitones for experiments spanning 0.5 semitones. We verify the appropriateness of these bandwidths by computing Monte Carlo split-half correlations (1,000 replicates) for two reference datasets (Study 1: harmonic dyads; Study 4B: major 3rd with 3 dB/octave roll-off). The results indicate excellent reliability in both cases (harmonic dyads: $r = 0.87$, 95% confidence intervals = [0.74, 0.94]; major third: $r = 0.93$, 95% confidence intervals = [0.82, 0.99]). We compute 95% confidence intervals for the smoothed ratings through nonparametric bootstrapping over participants, an approach which is robust to violations of standard parametric assumptions such as normality or homogeneity of variance; for computational tractability, we approximate these by computing bootstrapped standard errors ($N = 1,000$ replicates) and multiplying by 1.96 (Gaussian approximation). To facilitate interpretation, we also estimate peaks of the consonance curves using a custom peak-picking

algorithm, and compute confidence intervals for these peaks using the same bootstrapping algorithm (see details below).

**GSP.** Dense rating paradigms scale poorly to higher dimensions on account of the *curse of dimensionality* (e.g. ref. 75), which makes exhaustive sampling of the stimulus space impractical. For studying consonance in *triads* (chords comprising three tones), we therefore use Gibbs Sampling with People (GSP), a recent technique designed to tackle this dimensionality problem[40]. We parametrize the space of triads as pairs of intervals, building cumulatively from the bass note, meaning that (for example) a major triad is represented as the tuple (4, 3). This space also includes non-integer semitone chords such as (3.12, 4.17) and (4.12, 5.17). The space of possible triads can then be represented as a two-dimensional (2D) plane. Each trial begins at a point on this plane, with the participant being presented with a slider corresponding to either horizontal or vertical motion in the plane. Moving and releasing the slider triggers a new chord to be played corresponding to the updated position in the plane. The participant is told to move the slider to maximize the chord's consonance (Fig. 9a); when they are finished, the chord is then passed to next participant, who then manipulates the other interval and passes the chord to the next participant, and so on for a chosen number of iterations (typically 40) (Fig. 9b). This process can be modeled as a *Gibbs Sampling* process, well-known in computational statistics (Harrison et al.[40]). Following this model, we can estimate consonance for the full 2D space by applying a kernel density estimator (KDE) over the locations of the different trials generated by the process with a fixed bandwidth of 0.375. As before, the kernel bandwidth parametrizes a trade-off between bias and variance; here we chose a bandwidth of 0.375 semitones, and verified it as before with a Monte Carlo split-half correlation analysis (1,000 replicates), which demonstrated excellent reliability ($r = 0.93$, 95% confidence intervals = [0.91, 0.96]).

**Software implementation.** All experiments were implemented using PsyNet (v3.0.0; https://www.psynet.dev), our open-source framework for complex experiment design[40]. This framework builds on Dallinger (v7.6.0; https://github.com/Dallinger/Dallinger), a platform for experiment hosting and deployment (Supplementary Fig. 12). Participants engage with the experiment through a front-end interface displayed in the web browser, which communicates with a back-end Python server cluster that organizes the experiment timeline. The cluster is managed by the web service Heroku which orchestrates a collection of virtual instances that share the experiment management workload, as well as an encrypted Postgres database instance for data storage. See "Code availability" section for details regarding accessing the code for the implemented experiments.

**Stimuli**
Each of our chords can be expressed as a collection of absolute pitches. We express absolute pitches using MIDI notation, which maps each note of the 12-tone piano keyboard to a positive integer. Concert A (A4, 440 Hz) is by convention mapped to the value 69. Adjacent integers in this scale then correspond to adjacent semitones. Formally, the mapping is expressed as follows:

$$f = 440 \times 2^{(p-69)/12} \tag{1}$$

where $p$ is the MIDI pitch and $f$ is a frequency measured in Hz. An equal-tempered major triad rooted on middle C can then be expressed as the following tuple: [60, 64, 67]. This notation approach is useful for capturing the logarithmic nature of pitch perception[76].

In our experiments we also give chords intervallic representations. Intervallic representations are common in consonance experiments because consonance judgments are relatively insensitive to small-to-moderate changes in absolute pitch (though see ref. 77 for experiments exploring large changes). The intervallic representation expresses each non-bass tone as a pitch interval from the tone immediately below. Since there is no tone below the bass tone, this tone is omitted from the intervallic representation. Our experiments randomize over (dense rating) or manipulate (GSP) the intervallic representation; the bass tone is then treated as a separate parameter that is randomly sampled from a finite range on a trial-by-trial basis. For example, the dense rating procedure might generate the intervallic representation [4.1, 2.9]; in a given trial, this will be converted to an absolute representation of the form $[p_0, p_0 + 4.1, p_0 + 4.1 + 2.9]$, where $p_0$ is the randomly generated MIDI pitch of the bass tone.

In all experiments the bass tone was randomized by sampling from a uniform distribution over the MIDI pitch range 55–65 (G3–F4, 196–349 Hz). We performed this randomization to discourage participants from perceiving systematic tonal relationships between adjacent stimuli. The chord's pitch intervals were then constrained within a particular range, depending on the experiment: [0.5, 8.5] in Study 5, [0, 15] in Studies 1–3 and 4A, and $[I_c - 0.25, I_c + 0.25]$ with $I_c = 3.9, 8.9, 12$ in Study 4B.

We synthesized stimuli using Tone.js (v14.7.77; https://tonejs.github.io/), a Javascript library for sound synthesis in the web browser. Details of stimulus synthesis are provided below, split into the different tone types used in our experiments (see also Supplementary Tables 1 and 2). Most of our experiments used tones generated through additive synthesis. Each tone generated through additive synthesis can be expressed in the following form:

$$s(t) = A \sum_{i=0}^{n_H - 1} w_i \sin(2\pi f_i t) \tag{2}$$

where $s(t)$ is the instantaneous amplitude and $t$ is time. Different choices of weights $w_i$ and frequencies $f_i$ correspond to different tone spectra types. The parameter $A$ represents the overall amplitude.

Type I: Harmonic tones comprise $n_H = 10$ harmonic partials with $\rho$ dB/octave roll off. Concretely, $f_i = f_0 \times (i+1)$ and $w_i = 10^{-\omega_i/20}$ where $\omega_i = \rho \times \log_2(i+1)$ for $i = 0, \ldots, n_H - 1$.

Type II: Stretched and compressed tones are identical to harmonic tones except that $f_i = f_0 \times \gamma^{\log_2(i+1)}$ where $\gamma = 2.1, 1.9$ for stretched and compressed tones respectively. When $\gamma = 2$ we recover standard harmonic tones.

Type III: Pure tones comprise a single frequency ($n_H = 1$), $f_0$ and $w_0 = 1$.

Type IV: Complex tones with/without a 3rd harmonic comprise $n_H = 5$ harmonic partials with zero spectral roll-off. Formally, we write $f_i = f_0 \times (i+1)$ and $w_i = 1$ (type IV+) or $w_{i\neq2} = 1$ and $w_2 = 0$ for $i = 0, \ldots, 4$ (type IV-).

Type V: Bonang tones correspond to a synthetic approximation to a bonang tone, after Sethares[15]. Each tone comprises a custom complex tone given by $(f_0, 1.52f_0, 3.46f_0, 3.92f_0)$ with $w_i = 1$.

All additively synthesized timbres were presented with an ADSR envelope, comprising a linear attack segment lasting 200 ms reaching an amplitude of 1.0, a 100 ms exponential decay down to an amplitude of 0.8, a 30 ms sustain portion, and finally an exponential release portion lasting 1 s.

Some experiments additionally used a collection of more complex tones:

Type VI: Naturalistic instrument tones were based on samples from the Midi.js Soundfont database (https://github.com/gleitz/midi-js-soundfonts). The original database only provides samples for integer pitch values (i.e., 12-tone equal temperament); we therefore used the Sampler tool in the Tone.js library to interpolate between these values, synthesizing arbitrary pitches by pitch-shifting the nearest sample to the required frequency. By allowing our bass tones to rove over non-integer pitch values, we ensured that integer pitch intervals

were no more or less likely to exhibit pitch-shifting artifacts than non-integer intervals.

## Procedure
### Dense rating
**Task.** After completing a consent form and passing the pre-screening test, participants received the following instructions:

> In each trial of this experiment you will be presented with a word and a sound. Your task will be to judge how well the sound matches the word.

> You will have seven response options, ranging from Completely Disagree to Completely Agree. Choose the one you think is most appropriate.

This was then followed by a prompt informing participants of the response quality bonus (for further details, see Performance incentives). The experiment then proceeded as follows. In each trial, participants heard a sound (e.g., a dyad) and were presented with the following prompt:

> How well does the sound match the following word (pay attention to subtle differences): pleasant.

Participants then delivered their judgments on a rating scale (Supplementary Fig. 13) ranging from 1 (Completely Disagree) to 7 (Completely Agree). Participants were assigned randomly to stimuli and the stimuli themselves were sampled uniformly from the stimulus space. We note that while we did not collect ratings for adjectives other than pleasant in this study, the idea here was to use a wording that would allow the dense rating paradigm to be consistently applied across multiple adjectives in future work (as in Harrison et al.[40]).

### Gibbs sampling with people
**Task.** After completing a consent form and passing the pre-screening test, participants received the following instructions:

> In this experiment, you will listen to sounds by moving a slider. You will be asked to pick the sound which best represents the property in question.

To familiarize participants with the interface, each participant had to complete three training examples prior to the start of the actual experiment. These were presented with the following instructions:

> We will now play some training examples to help you understand the format of the experiment. To be able to submit a response you must explore at least three different locations of the slider.

The training examples took an identical format to the main experiment trials, but with their intervals generated randomly rather than through a GSP process. Moreover, prior to the training trials participants were informed of the performance quality bonus (for further details, see Performance incentives).

After completing the training phase, the main experiment began. Participants received the following instructions:

> The actual experiment will begin now. Pay careful attention to the various sounds! Sometimes the differences between the sounds can be subtle, choose what seems most accurate to you. Remember: the best strategy is just to honestly report what you think sounds better! You must explore at least three locations of the slider before submitting a response. Also, sometimes the sounds might take a moment to load, so please be patient.

The experiment then proceeded as follows: in each trial, the participant was assigned to one of the available GSP chains (Supplementary Fig. 14), and was provided a stimulus (e.g., a chord) and a slider (Supplementary Fig. 15). The slider was coupled to a particular dimension of the stimulus space (e.g., an interval) and changed from trial to trial. The participant was presented with the following prompt:

> Adjust the slider to match the following word as well as possible: pleasant.

> Please pay attention to the subtle differences.

In other words, the participant was instructed to explore the slider to find the sound that was most associated with the word in question. The resulting stimulus was then passed along the GSP chain to the next participant, with each successive participant optimizing a different dimension. To circumvent any potential biases toward left or right slider directions, the direction of the slider was randomized in each trial so that in approximately half of the trials the right of the slider corresponded to bigger values, and in the other half it corresponded to smaller values.

**Starting values.** The starting location of each GSP chain was sampled from uniform distributions over the permissible ranges of the active stimulus dimensions.

**Assigning participants to chains.** The GSP process involves constructing chains of trials from multiple participants. We achieved this by applying the following process each time a participant was ready to take a new trial:
- Find all chains in the experiment that satisfy the following conditions:

  - The chain is not full (i.e., it has not yet reached the maximum allowed number of iterations);
  - The participant has not already participated in that chain;
  - The chain is not waiting for a response from any other participant.
- Randomly assign the participant to one of these chains.

**Participant timeouts.** Sometimes a participant will unexpectedly stop participating in the experiment. In order to prevent chains being blocked by perpetually waiting for such participants, we implemented a timeout parameter, set to 60 s, after which point the chain would stop waiting for the participant and instead open itself up to new participants. If the blocking participant did eventually submit a trial, they would be allowed to continue the experiment, but their trial would not contribute to the GSP chain.

**Headphone screening test.** To ensure high-quality listening conditions we used a previously validated headphone screening test[78]. Each trial comprises a three-alternative forced-choice task where the participant is played three tones and asked to identify the quietest. The tones are designed to induce a phase cancellation effect, such that when played on loudspeakers the order of their quietness is altered, causing the participant to respond incorrectly. To pass the test the participant had to answer at least four out of six trials correctly. As well as selecting for headphone use, this task also helps to screen out automated scripts (bots) masquerading as participants[79], since successful completion of the task requires following written instructions and responding to auditory stimuli.

**Performance incentives.** Although our tasks are subjective in nature, meaning that there are no a priori right or wrong answers, we

nevertheless wanted participants to listen carefully and perform the task honestly. To do that, prior to starting the main experiment, participants received one of the following instructions:

> The quality of your responses will be automatically monitored, and you will receive a bonus at the end of the experiment in proportion to your quality score. The best way to achieve a high score is to concentrate and give each trial your best attempt. (Dense rating task)

> The quality of your responses will be checked automatically, and high quality responses will be bonused accordingly at the end of the experiment. The best strategy is to perform the task honestly and carefully, and then you will tend to give good results. (GSP task)

We purposely did not tell participants exactly how these quality scores were computed, so as to avoid biasing the participants to answer in a particular way (for example answering in a way that matches the responses of other participants). In actuality, we computed these scores by quantifying the participant's self-consistency, reasoning that participants who take the task carefully are likely to provide similar responses when presented with the same trial multiple times. This seemed the most reasonable option given the lack of ground truth for subjective tasks such as these.

Self-consistency was estimated as follows. Upon completion of the main experiment trials, participants received a small number of trials (three for GSP, and five for dense rating) which repeated randomly selected trials that were encountered earlier. The data from these trials contributed only to consistency estimation and not to the construction of the main experiment. Consistency was quantified by taking the Spearman correlation between the two sets of numerical answers. Participants were not informed of their consistency score, but at the end of the experiment they received a small monetary bonus in proportion to their score, constrained to range between 0 to 0.5 dollars. The exact mapping between score and bonus was $\min(\max(0, 0.5 \times \text{score}), 0.5)$ for dense rating and $\min(\max(0, \text{score} - 0.5), 0.5)/2$ for GSP.

## Participants
The US cohort for Studies 1–5 was recruited from Amazon Mechanical Turk (AMT), a well-established online crowd-sourcing platform. We specified the following recruitment criteria: that participants must be at least 18 years of age, that they reside in the United States, and that they have a 95% or higher approval rate on previous AMT tasks. All participants provided informed consent in accordance with the Max Planck Society Ethics Council approved protocol 2020_05; all data collection was anonymous in order to protect participants' privacy. While we only allowed each individual to participate once in a given experiment, we did not regulate or track whether individuals participated in multiple experiments. Our reported AMT participant numbers therefore correspond to the total number of times someone participated in our experiments, rather than the total number of unique individuals across experiments.

We ran each AMT experiment for about a day, targeting about 150–200 participants for 1D experiments, and targeting somewhat larger cohorts (~200–350) for the multi-dimensional and tuning experiments; the latter experiments required exploring larger stimulus spaces and/or more subtle perceptual effects. These target numbers were established via pilot experiments and verified post-hoc using Monte Carlo split-half reliability analyses, which indicated an excellent reliability for both the dyad paradigm ($r = 0.87$, 95% confidence intervals = [0.74, 0.94]) and the triad paradigm ($r = 0.93$, 95% confidence intervals = [0.91, 0.96]). All together the AMT cohort comprised 4204 verified participants, in addition to 2373 participants who failed to complete the pre-screening tasks or the main experimental tasks. Details of the number of unique participants

within each individual experiment are provided in Supplementary Tables 1 and 2.

Three additional experiments (Study 2B) were completed by a cohort of South Korean participants which were recruited through a research assistant residing in the local area[80]; AMT was not possible in this case as AMT does not currently run in South Korea. Participants were required both to be born in South Korea and to be current residents there. In order to maximize the amount of available data, each participant was allowed to participate in the same experiment up to three times. Each participant provided informed consent following the Max Planck Society Ethics Council approved protocol 2021_42, and took a Korean-language version as translated by a native speaker.

The overall compensation for participating in each experiment was computed by estimating the total duration of the experiment and multiplying by a rate of $9/h. Individual participants were paid in proportion to the amount of the experiment that they completed. Participants were still compensated even if they left the experiment early on account of failing a pre-screening task.

## Questionnaire
Upon completion of each experiment we collected demographic information as well as years of musical experience from participants ("Have you ever played a musical instrument? If yes, for how many years did you play?"). In the US cohort, reported ages varied in the range 18–81 ($M = 37.5$, SD = 11.0), and 39.8% self-identified as female, 59.0% as male, and 1.2% as other. We note that sex and gender were not part of our study design as we were interested in the overall population level effects of timbre on consonance perception. Participants self-reported 0–55 ($Med = 2$, $M = 4$, SD = 6.6) years of musical experience. In the South Korean cohort, the reported age statistics were ($M = 27$, SD = 10.50), 50% self-reported as female and 50% as male, and those of the years of musical experience were ($Med = 1$, $M = 2.19$, SD = 2.67). Details of the demographic information of each individual experiment are provided in Supplementary Tables 1 and 2.

## Ethics and inclusion statement
The present study has been conducted in compliance with approved Max Planck Society Ethics Council protocols (2020_05 and 2021_42). The former protocol (2020_05) covered recruitment in the US, and the latter protocol (2021_42) covered recruitment in South Korea. All participants provided informed consent prior to participation in the study. To ensure that our research conforms to ethical standards and local norms, we included researchers who are native and have been raised in English and Korean countries (Author P. M. C. Harrison born in the UK and is a native English speaker; Author H. Lee born in South Korea and is a native Korean speaker). We did not obtain additional ethics approval from a local South Korean institution as participation was conducted remotely, a native researcher was involved, and our approved protocol (2021_42) covered remote data collection. Participants were treated fairly and transparently as much as possible. Our research did not involve stigmatization, incrimination, or discrimination, and none of the experiments posed any risk to participants.

## Individual experiments
### Dense rating
**Studies 1, 2, 4.** These experiments elicited pleasantness judgments for dyads of various timbres. Interval sizes were randomly sampled from uniform distributions, with the ranges of these distributions varying between studies: Studies 1 and 2 used a range of [0, 15] semitones, whereas Study 4 used ranges of the form $[I_c - 0.25, I_c + 0.25]$ where $I_c = 3.9, 8.9, 12$ respectively. The $I_c$ values correspond to averages of just-intoned and equal-tempered tuning values at the major third, sixth and octave, rounded to one decimal place. In principle, each stimulus was to receive exactly one rating; occasionally for technical reasons the

same stimulus was nonetheless administered to more than one participant, but then the latter response was excluded from the data analysis. The exact number of participants, timbres and average number of ratings per participant and per stimulus are summarized in Supplementary Table 1.

**Study 3**. This experiment elicited pleasantness judgments from 322 participants for dyads with Type I timbre and varying spectral roll off ($\rho$). Stimuli were sampled uniformly from $[0, 15] \times [0, 15]$ where the first range corresponds to interval size in semitones and the second range corresponds to spectral roll-off in dB/octave.

**GSP**

**Study 5**. In each experiment participants completed a collection of 200 GSP chains each of length 40 (excluding the random seed). The stimuli consisted of triads composed with different fixed timbres and parametrized by two intervals in the range [0.5, 8.5] semitones. A summary for each individual experiment can be found in Supplementary Table 2.

**Data analysis and visualization**

Each of our behavioral experiments involves sampling from some kind of continuous space. In each case we apply some kind of nonparametric smoothing to infer a smooth consonance terrain from these discrete samples. The precise nature of this smoothing varies depending on the paradigm type (rating versus GSP), the dimensions of the stimulus space, and the nature of the dimension (pitch interval versus timbre).

Throughout our paper we use bootstrapping for our inferential statistics, partly because of its well-known robustness to assumption violations, and partly because it can be applied to complex analysis methods like ours for which orthodox inferential statistics are not available.

Because of the bootstrapping procedure we did not rely on $p$ values in the arguments in the paper and thus did not report it. However, the provided 95% confidence intervals can be used for a similar purpose of evaluating whether the results are compatible with alternative hypotheses.

**1D rating experiments (Studies 1, 2, 4)**. We computed the 1D consonance profiles over a grid of 1000 points spanning the interval range of interest. In Studies 1, 2, and 4A, the range spanned 15 semitones; in Study 4B, the range spanned 0.5 semitones.

The behavioral profiles were computed by taking the trial-level rating data, $z$-scoring the ratings within participants, and then applying a Gaussian kernel smoother. For experiments spanning 0–15 semitones, we used a kernel with standard deviation of 0.2 semitones; for experiments spanning 0.5 semitones, we used a kernel with standard deviation of 0.035 semitones.

The interference model profiles were computed directly from the corresponding computational models, supposing a bass note corresponding to C4 (~262 Hz), and modeling the tone spectra on the corresponding experimental stimuli (see below for more details on the models). We smoothed all model profiles using the same process as for the behavioral profiles to maximize comparability between the two sets of profiles.

We computed confidence intervals for the behavioral profiles by nonparametric bootstrapping over participants. To keep computation time tractable we used 1000 bootstrap replicates to estimate the standard error and then estimated 95% confidence intervals by making a Gaussian approximation ([mean − 1.96 × SE, mean + 1.96 × SE]). We used this same approach for all bootstrapped analyses in the paper.

We computed peaks for the behavioral profiles by applying a custom peak-picking algorithm comprising the following steps:

(1) Take the kernel-smoothed behavioral profile as an input; this corresponds to a vector of intervals and a vector of corresponding pleasantness values, both of length 1,000.
(2) Approximate this profile using a cubic smoothing spline (implemented as smooth.spline in R). We set the equivalent number of degrees of freedom to 100 to ensure perfect interpolation. Write this spline function as $f(x)$, where $x$ is the interval in semitones.
(3) Compute the first and second derivatives of this spline function ($f'(x), f''(x)$).
(4) Compute the range of the spline function, writing it as range$(f(x))$.
(5) Compile a list of all *peaks* in the function. A peak is defined as a value $x_i$ where the following holds: $f'(x_i) > 0 > f'(x_{i+1})$ and $f''(x_i) < -\text{range}(f(x))/20$.
(6) Compile a list of all *troughs* in the function. A trough is defined as a value $x_i$ where the following holds: $f'(x_i) < 0 < f'(x)_{i+1}$ and $f''(x_i) > \text{range}(x)/20$.
(7) Merge peaks that aren't separated by deep enough troughs, in each case keeping the tallest peak. Formally: write $P_i$ for the location of the ith peak, $P_{i+1}$ for the next peak, and $T_{min}$ for the lowest trough in between; we consider $T_i$ to be deep enough if $\min(f(P_i), f(P_{i+1})) - f(T_{min}) \geq \alpha \times \text{range}(f(x))$ where $\alpha = 0.01$.
(8) Discard peaks that aren't sufficiently *sharp*. A peak $P$ is considered sharp if it satisfies both of the following conditions:

$$\exists a \in [P - 0.5, P] : f(P) - f(a) \geq \beta \times \text{range}(f(x)) \quad (3)$$

$$\exists b \in [P, P + 0.5] : f(P) - f(b) \geq \beta \times \text{range}(f(x)) \quad (4)$$

where $\beta = 0.01$.

We estimated the reliability of these peaks via the same nonparametric bootstrapping procedure described above. We took the 1000 bootstrap replicates of the kernel-smoothed behavioral profiles created previously, and ran the peak-picking algorithm on each of these, producing 1000 sets of peak estimates. For each of the peaks obtained from the original behavioral profile, we defined a neighborhood as being within ±0.5 semitones of that peak, and counted the proportion of bootstrap iterations where a peak was observed within that neighborhood. If this proportion was greater than 95%, we considered that peak to be statistically reliable. We then computed 95% confidence intervals for the location of that peak by taking the standard deviation of those bootstrapped peak locations (i.e., the bootstrapped standard error), and then performing the same Gaussian approximation described above ([mean − 1.96 × SE, mean + 1.96 × SE]).

**2D rating × roll-off experiment (Study 3)**. We computed consonance profiles for Study 3 by factorially combining three spectral roll-off levels (2, 7, and 12 dB/octave) with the same 1000-point interval grid from the previous analyses. We smoothed the behavioral data using an analogous kernel smoothing process to before, using a Gaussian function with an interval standard deviation of 0.2 semitones (as before) and a roll-off standard deviation of 1.5 dB/octave. All other aspects of the analysis (smoothing the harmonicity models, estimating peak locations, and computing confidence intervals) were identical to before.

**2D interval × interval experiment (Study 5)**. We computed consonance profiles for Study 5 over a grid of 500 × 500 points spanning 0.5–8.5 semitones on both dimensions. Following the standard GSP approach[40], the behavioral consonance profiles correspond to kernel density estimates over populations of trials. Here we used a kernel density estimator with a bandwidth of 0.375 semitones. We computed consonance model outputs assuming a bass note of C4 as before, and smoothed the outputs using a bandwidth of 0.375/2 = 0.1875.

**Spectral approximations for naturalistic instruments.** Study 1B reports spectral approximations for several naturalistic musical instruments (flute, guitar, piano). We calculated these spectra using the method of McDermott et al.[81] as applied to audio samples for a C4 tone. First the signal was passed through an array of equally spaced cosine filters expressed on the MIDI scale, with 0.5 semitone spacing between filters, 50% overlap between adjacent filters, and filter locations spanning the range 15–119 semitones. Temporal envelopes at each of the filter locations were then extracted by taking the modulus of each filter's analytic signal as computed using the Hilbert transform, then applying a non-linearity ($f(s) = s^{0.3}$), and then resampling to 400 Hz. Temporal envelopes for the first 20 harmonics were then estimated by reversing the non-linearity (i.e., applying $g(s) = s^{1/0.3}$) and evaluating a Gaussian kernel smoother ($\sigma = 1.598$ Hz) at each harmonic frequency over the entire temporal trajectory. Each harmonic's temporal envelope was then averaged to produce a single amplitude score.

## Consonance models

While many psychoacoustic accounts of consonance perception have been presented over the centuries, recent literature has converged on two main candidate explanations: (1) interference between partials, and (2) harmonicity (see ref. 20 for a review). We address both accounts in this paper using computational modeling, as described below.

**Interference models.** According to interference accounts, consonance reflects interference between the partials in the chord's frequency spectrum. The nature of this interference depends on the distance between the partials. Distant partials and very close partials elicit minimal interference; however, partials separated by a moderately small distance (of the order of a semitone) elicit a large amount of unpleasant interference. This interference is most commonly thought to derive from fast amplitude fluctuation (beating[26]) but may also reflect masking[8,82].

The literature contains many interference-based consonance models (see ref. 20 for a review). These models vary in their mechanistic complexity, but interestingly the older and simpler models seem to perform better on current empirical data[20]. Here we use a collection of so-called pure-dyad interference models[31,41,43] which calculate interference by summing contributions from all pairs of partials in the acoustic spectrum, where each pairwise contribution is calculated as an idealized function of the partial amplitudes and the frequency distance between the partials. We focus particularly on the model of Hutchinson and Knopoff[41], which performed the best of all 21 models evaluated in Harrison and Pearce[20], but we also explore the other models in the Supplementary Information. We avoided testing more complex waveform-based models (e.g. refs. 83,84) because of their high computational demands and relatively low predictive performance[20].

At the core of the Hutchinson-Knopoff model is a dissonance curve that specifies the relative interference between two partials as a function of their frequency distance, expressed in units of critical bandwidths (Supplementary Fig. 11). This relative interference is converted into absolute interference by multiplying with the product of the amplitudes of the two partials. The main differences between the Hutchison-Knopoff, Sethares, and Vassilakis models correspond to the precise shapes of the dissonance curves and the precise nature of the amplitude weighting.

The original presentation of the Hutchinson-Knopoff model defined the dissonance curve solely in graphical form. Here we use a parametric approximation of this curve introduced by Bigand et al.[50]: $D(x) = (4x \exp(1 - 4x))^2$, where $D(x)$ is the dissonance contribution of a pair of partials separated by critical bandwidth distance $x$ (Supplementary Fig. 11, top panel). They model critical bandwidth distance ($x$)

as a function of the frequencies of the two partials $f_1, f_2$:

$$x = |f_2 - f_1| / 1.72 \left(\frac{f_1 + f_2}{2}\right)^{0.65} \qquad (5)$$

Our revised model (as included in the composite model plots) includes several additional changes motivated by the results of our experiments. These changes involve new model parameters ($p$, $q$, and $r$) which were optimized numerically following the steps described in Parameter optimization (see below).

First, the dissonance curve $D(x)$ is revised to incorporate a preference for slow beats (Supplementary Fig. 11, bottom panel):

$$
\begin{aligned}
D^*(x) &= \frac{x}{p} D(x) - q\left(1 - \frac{x}{p}\right)\left(1 + \sin(2\pi \frac{x}{p} - \frac{\pi}{2})\right), \quad \text{for } x < p \\
&= D(x), \qquad\qquad\qquad\qquad\qquad\qquad\qquad \text{otherwise}.
\end{aligned}
\qquad (6)
$$

where $p = 0.096$ is the slow-beat boundary (the distance at which the pleasantness of slow beats starts contributing) and $q = 1.632$ is the slow-beat pleasantness (the strength of the slow-beat pleasantness effect).

The original Hutchinson-Knopoff model sums together dissonance contributions from all pairs of partials, weighting each contribution by the product of those partials' amplitudes. The consequence is that dissonance is proportional to sound *intensity*.

$$\text{Dissonance (original)} = \sum_{i,j:i<j} A_i A_j D(f_i f_j) / \sum_k A_k^2 \qquad (7)$$

In our revised model, we add an additional parameter $r$ that nuances this amplitude weighting. Setting $r = 2$ recovers the intensity-weighting of the original model. Setting $r = 1$ makes dissonance proportional to amplitude rather than intensity. The optimized value of $r = 1.359$ sits somewhere between these two interpretations:

$$\text{Dissonance (new)} = \sum_{i,j:i<j} (A_i A_j)^{r/2} D^*(f_i f_j) / \sum_k A_k^r \qquad (8)$$

**Harmonicity models.** According to harmonicity accounts, consonance is grounded in the mechanisms of pitch perception. Pitch perception involves combining multiple related spectral components into a unitary perceptual image, a process thought to be accomplished either by template-matching in the spectral domain or autocorrelation in the temporal domain (see ref. 85 for a review). Consonance perception can then be modeled in terms of how well a particular chord supports these pitch perception processes. Here we test three such models: two based on template-matching[42,45] and one based on autocorrelation, after Boersma[44] (see below for details). We focus particularly on the model of Harrison and Pearce[42] because of its high performance in Harrison and Pearce[20], but we also explore the other models in the Supplementary Information. We excluded several other candidate models because they are insensitive to spectral manipulations, the main focus of this paper[5,17,86].

**The Harrison-Pearce and Milne models.** Following Milne[45], the Harrison-Pearce model uses a harmonic template corresponding to an idealized harmonic complex tone. The template is expressed in the pitch-class domain, a form of pitch notation where pitches separated by integer numbers of octaves are labeled with the same pitch class. It can be transposed to represent different candidate pitches; Supplementary Fig. 16a shows templates for C, D, E, and F.

Each input chord is likewise expressed as an idealized spectrum in the pitch-class domain, after Milne[45] (Supplementary Fig. 16b). This involves expanding each chord tone into its implied harmonics, making sure to capture any available information about the strength of the harmonics (e.g., spectral roll-off) and their location (e.g., stretched versus non-stretched).

A profile of virtual pitch strength is then created by calculating the cosine similarity between the chord's spectrum and different transpositions of the harmonic template, after Milne[45] (Supplementary Fig. 16b). For example, the virtual pitch strength at 2 corresponds to the cosine similarity between the chord's spectrum and a harmonic template with a pitch class of 2 (i.e., a D pitch template).

Finally, harmonicity is estimated as a summary statistic of the virtual pitch strength profile. The Harrison-Pearce model treats this profile as a probability distribution, and computes the information-theoretic uncertainty of this distribution, equivalent to the Kullback-Leibler divergence to this distribution from a uniform distribution; high uncertainty means an unclear pitch and hence low harmonicity. Milne's[45] model takes the same approach, but instead returns the height of the highest peak of this distribution.

**The autocorrelation model.** The autocorrelation model uses the fundamental-frequency estimator of Boersma[44], as implemented in the Praat software and accessed via the Parselmouth package[87]. The algorithm works by looking for the maximum of the sound's autocorrelation function (i.e., the temporal interval at which the sound correlates maximally with itself).

The following steps were used to estimate the harmonicity of a given chord:
(1) Synthesize the chord to an audio file using additive synthesis;
(2) Estimate fundamental frequency from the audio file using Boersma's[44] algorithm, using a time-step of 0.1 s, and bounding candidate fundamental frequencies to lie above 10 Hz but more than four semitones below the lowest chord tone.
(3) Take the median of these fundamental frequency estimates. After Stolzenburg[17], high fundamental frequencies are taken as implying high periodicity and hence high harmonicity.

**Composite model.** Our composite model combines both interference[41] and harmonicity[42] models, including the modifications described above. The two models are combined additively with a weight of −1 for the interference model and +0.837 for the harmonicity model (see Parameter optimization for details). When applying the composite model we also median-normalize the outputs within each experiment (i.e., subtracting the median value across the whole experiment from each model output), reflecting the way in which participants calibrate their scale usage to the stimuli within each experiment. Note that we plot the final version of the composite model throughout the paper, rather than plotting incremental versions as motivated by each experiment, to ensure that later model changes do not spoil predictions for earlier experiments.

**Parameter optimization.** Our new models include several key parameters, listed in Supplementary Table 3. We set initial versions of these parameters based on a combination of theory and manual exploration of the data; we then numerically optimized these parameters using a gradient-free optimizer as described below.

Optimizing model parameters requires defining an objective function that is not too time-consuming to compute. We therefore excluded the triad experiments from the objective function, and solely modeled data from the dyad experiments. For each dyad experiment, we computed consonance profiles for models and participants as described earlier. We found that correlation metrics were a poor measure of qualitative fit between models and participants, so we instead measured fit by comparing the peaks between the model and the participant profiles. Participant profile peaks were computed as previously; model profile peaks were obtained analogously, but instead of relying on

bootstrapping to remove unreliable peaks (the model outputs are deterministic and hence can't be bootstrapped) we filtered excess peaks by increasing the peak-finding minimum depth parameter ($\beta$) from 0.01 to 0.05. We then defined model fit by looking at the overlap between participant peaks and model peaks. In particular, we characterized a pair of participant and model peaks as overlapping if they were separated by less than 2.67% of the overall interval range in that study (i.e., 0.4 semitones in the 15-semitone experiments, or 0.013 semitones in the 0.5-semitone experiments), and then computed the Jaccard similarity between the sets of participant and model peaks. The overall model fit was then calculated as the mean Jaccard similarity across all the dyad experiments.

We optimized this objective function as a function of the parameters listed in Supplementary Table 3 using the subplex algorithm[88] as implemented in the NLopt package (http://github.com/stevengj/nlopt), and using the parameter bounds specified in Supplementary Table 3. The model converged after 245 iterations to a mean Jaccard similarity of 0.467. The resulting model parameters (Supplementary Table 3) are used for all visualizations and analyses involving the composite model.

### Reporting summary
Further information on research design is available in the Nature Portfolio Reporting Summary linked to this article.

### Data availability
All data generated in this work have been deposited in a publicly available OSF repository under the following URL: https://osf.io/83w2b/ (DOI: 10.17605/OSF.IO/83W2B). To further facilitate the process, data can also be explored and exported via the following interactive web app: https://pmcharrison.gitlab.io/timbre-and-consonance-paper/supplementary.html. Naturalistic instrument tone samples used in this study are available at the Midi.js Soundfont database: https://github.com/gleitz/midi-js-soundfonts. Benchmark behavioral datasets for Study 1[38,39,46] are available through the following repository: https://gitlab.com/pmcharrison/timbre-and-consonance-paper.

### Code availability
All necessary code for replicating our analyses, and reproducing our experiments are deposited at the following OSF repository: https://osf.io/83w2b/ (DOI: 10.17605/OSF.IO/83W2B). These can also be accessed individually through the following links: Individual experimental implementations, and experiment templates: https://gitlab.com/raja.marjieh/consonance-and-timbre-data. Analysis code: https://gitlab.com/pmcharrison/timbre-and-consonance-paper. Interactive web app for exploring data and models: https://pmcharrison.gitlab.io/timbre-and-consonance-paper/supplementary.html.

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

## Acknowledgements
FD was supported by an Erasmus+ scholarship.

## Author contributions
Authors RM, PH and NJ designed the experiments. Author RM conducted the experiments with US participants (Studies 1–5); authors HL and FD conducted the experiments with Korean participants (Study 2B). Author PH performed the psychological modeling and prepared the Supplementary Materials. Authors RM, PH and author NJ wrote the paper. All authors edited and commented on the manuscript.

## Funding

## Competing interests
The authors declare no competing interests.
