## [Peer Review File · Nature Communications]

Timbral effects on consonance disentangle psychoacoustic mechanisms and suggest perceptual origins for musical scalesReviewers' Comments:

Reviewer #1:

Remarks to the Author:

The manuscript presents five large-scale online experiments investigating the influence of a sound's spectral content on subjective judgments of pleasantness. Pleasantness judgments are used as a proxy for the musical notion of consonance. The logic in the series of experiments is to test two classic models of consonance: an "interference" model, based on the theory of roughness introduced by Helmholtz, and the "harmonicity" model, with deeply established roots in music theory and more recent proposals by e.g. Terhardt. A "composite" model is also introduced, consisting of a simple linear combination of the predictions of the interference and harmonicity models. The experimental results show that the composite model is needed to capture effects of spectral content on pleasantness.

The question asked in the manuscript relates to perhaps one of the longest standing puzzles in (Western) music theory: what makes some combination of tones pleasant when sounded together, or con-sonant? As such, it will be of interest to the broad readership of the journal.

For full disclosure, I had already reviewed a previous version of the manuscript. In a nutshell, I think the way the study has been strengthened and enriched is very convincing. A strength of the manuscript is the sheer scale of the dataset: more than 4k participants overall. While the theoretical questions had already been framed in previous work, such as Sethares, 2005 and one of the authors' own review in 2020, the experimental data provided here are unique thanks to their scale and also thanks to some clever and original design choices. The composite model allows to make sense of the experimental results, providing a satisfactory resolution to seemingly contradictory findings favoring either interference or harmonicity. As an aside, I would also like to commend the authors for the very impressive Supplemental material they provide. In particular, the animated/audio version of the results would make for nice teaching materials or help non-musical readers to get the point of the study.

My only suggestions would revolve around how the composite model is introduced and specified, and whether it would be possible to get even more insight from it by considering inter-individual differences.

1) In terms of presentation, the models are all described in the Methods, with only a brief introduction in the main text. This makes for a more compact presentation of the experimental data, but it may also detract from the understanding of the insights provided by the modeling work. This is surely something to be left to the appreciation of the authors, but perhaps providing a bit more details on the models early on would be useful?

2) The parameters for the new models were fitted by hand, if I understood correctly (for instance, the improved way to take into account amplitude in Hutchinson & Knopoff, the way to introduce negative dissonance for slow beats, the relative weights of interference and harmonicity in the composite model). While the exercise has clearly been successful, I wonder if it would not be useful to optimize these parameters by performing a formal fit to the data.

3) If point 2) were to be implemented, then this would perhaps open up a very interesting extension to the modeling work. It seems clear from Fig. 1B that there is a large individual variability in the data. The Methods convincingly show how meaningful average patterns were extracted from the raw judgements, and then most of the descriptions and discussions in the manuscript focus on the average trends (with the exception of the musician/non-musician analyses). However, I wonder whether the authors would not be in an ideal position to also comment on how pleasantness judgments may systematically differ across participants. From the data, are there "clusters" of patterns of consonance? Could it be because listeners have different relative weights for interference and harmonicity in their very own composite model? I fully appreciate that the manuscript is already very

rich, so I'd happily accept a comment to the effect of "this will be left for future studies" if this represents too much of an undertaking.

Reviewer #2:

Remarks to the Author:

The paper is very clearly written and presents a novel and interesting experimental method, which uses finely grained microtonal tunings with harmonic and inharmonic spectra to decorrelate the potential influences of roughness and harmonicity on the pleasantness of musical dyads and triads. The findings are also very interesting – not only demonstrating the somewhat independent roles played by both mechanisms, but also suggesting some very interesting refinements to the classic roughness model; in particular, modifying the basic dissonance curve to account for the pleasantness of slow beats, which is a priori plausible and – for the first time – here experimentally demonstrated by using such finely grained pitch manipulations.

I strongly recommend this manuscript for publication due to the above-mentioned innovations in experiment method combined with interesting and novel findings related to human perception of harmony that these methodological innovations make possible, and a revised roughness model to account for these findings. The statistical and mathematical methods are well-conceived and rigorous. The methods and findings will certainly influence future investigations of the human experience of music.

I will now add a few minor suggestions. None of these need to be addressed, but the authors may find some of the comments useful.

p.2, l.60: "This would explain why scale systems across the world seem to have developed to favor harmonic pitch intervals" This is stated rather too definitively -- although common, it is not universally true (gamelan scales being a case in point)

p.3, l.89: "Previous consonance research has used stimuli drawn solely from discrete scales, in particular the Western 12-tone chromatic scale" This could be read as suggesting that all previous research has used only 12-TET, which of course is not true.

p.5, l.160: "Other possible synonyms exist (e.g., smoothness, purity, harmoniousness), but in practice human ratings tend to be highly correlated across these synonyms (Lahdelma & Eerola, 2020)" I wonder if this is slightly overstating things. There are important differences noted in the cited paper between some terms; in particular, how "tension" and "pleasantness" are, respectively, stable or change across differing musical familiarities and expertise. "Tension" is not mentioned in the parentheses nor other commonly used synonyms such as "stability".

p.26, l.669: "They also provide an empirical justification for certain practices in the experimental music tradition of 'Dynamic Tonality' (Plamondon et al., 2009)" I wonder if Sethares, W. A., Milne, A. J., Tiedje, S., Prechtl, A., and Plamondon, J. (2009). Spectral tools for Dynamic Tonality and audio morphing. *Computer Music Journal*, 33(2):71–84 would be a better or additional citation here?

p.29, l.726: "The preferences we document for slight inharmonicity moreover shed new light on the historical development of Western tuning systems: they indicate that the slight impurities of systems such as mean-tone and equal temperament may not detract from subjective pleasantness, as is typically assumed, but instead positively contribute to it." I think this is an interesting point and certainly applicable to several meantone tunings. But for 12-TET around middle-C, the beating of the lower tone's 5th harmonic and the upper tone's 4th will be about 10Hz and so somewhat above the range of pleasant slow beating. Historically speaking, meantone probably gradually adapted into 12-TET so these pleasant impurities at the start may have eased that path even though the end-point is

not really consistent with the slow-beating hypothesis.

p.36, l.949: It would be more conventional for \sin to be in an upright font; e.g., by using \sin in LaTeX

p.38, l.1001: "How well does the sound match the following word (pay attention to subtle differences): pleasant" What motivates this specific choice of wording? It suggests that there are likely to be other words than "pleasant" for the comparisons. Were there? If so they should be reported here even if not discussed further. If not, is it problematic that the participants may have been waiting for a different question to appear (which never did)? Is there an advantage to asking the question in this way rather than just providing a rating scale with something like "very unpleasant" and "very pleasant" at the two ends? It would be useful to provide some reasoning behind the wording question. (Overall this is a very minor comment, but I was struck by the unusual wording of the question.)

p.47, l.1245: "We smoothed all model profiles using the same process as for the behavioral profiles" This makes no sense to me. Why were the model's predictions smoothed? The purpose of smoothing is to reduce "high-frequency" noise so as to reveal an underlying distribution. But there is no stochastic element to the predictions of roughness or harmonicity made by any of these models, so I cannot see any justification for smoothing their results. Can this be explained? Or, if there is no explanation, perhaps the smoothing should not be applied?

p.55, l.1462: "The two models are combined additively with a weight of -1 for the interference model and +0.75 for the harmonicity model (weights chosen on the basis of manual exploration)." Excellent to see a single model (identical parameter values) applied to every data set (particularly given that this has been abused resulting in gross overfitting in some music cognition research, which will remain nameless for the purpose of this review!). However, I was surprised to see the weights had been set by "manual exploration", which I presume means changing the relative weights until the curves seem – by eye – to fit better. Given the statistical and mathematical rigour of most of the paper, couldn't there be a more systematic way to optimise this relative weight? Or is there a good argument for not employing a more systematic approach?

p.55, l.1467: "Note that we plot the final version of the composite model throughout the paper, rather than plotting incremental versions as motivated by each experiment, to ensure that later model changes do not spoil predictions for earlier experiments." Something similar but less complete is mentioned earlier in the paper. However, it wasn't until I read this part that I was sure whether the weights of the two components had been kept the same or just the adjustments related to using amplitude (instead of intensity) and the modification of the dissonance curve. I would suggest making it clear when first mentioned that the weights are also kept the same because this is an important positive aspect of this article's analyses.

Reviewed by Andrew Milne

Reviewer #3:

Remarks to the Author:

In this paper the authors perform a series of psychoacoustical online experiments in which various combinations of complex and pure tones are presented in order to explore the concept of dissonance and its dependence on various factors, including harmonicity of the constituent tones, spectral shape (roll-off), and presence/absence of certain harmonics. The methodology has some advantages (sample size, mostly, but also ability to sample across a wide parameter space); but it also has some weaknesses (e.g. mixing of between vs within-subject variance, which makes it hard to identify individual difference; as well as poor control over attention, sound level, hearing status, and other details that are normally highly controlled in traditional psychophysics). But for the purpose of providing a data over wide ranges of parameter combinations, as done here, it seems to work

remarkably well. The results demonstrate some interesting interactions, especially with regards to how harmonicity (stretching vs compressing the harmonics) influences judgments of consonance. Another important aspect of the study is that two models (harmonicity vs interference) are compared explicitly, which has rarely (or maybe never) been done in a formal way; plus they also develop their own hybrid model combining some features of each, which turns out to fit the data better than either of the ones most commonly espoused in the literature.

Despite the technical tour-de-force, and the paper's many good features, I miss the bigger impact that these findings might have. It is certainly of interest to a small segment of the scientific community (among whom I would count myself) interested in the psychoacoustics of consonance. But I failed to see the conceptual advance that would have implications for other domains of broader interest, including speech, or even music perception. In the case of music, the role of consonance/pleasantness in music is hugely modulated by context, but the approach taken here eschews all such considerations, which is fine for purposes of identifying the basic relations but not very useful for inferring how these relations actually play out in real music. I can see this paper fitting very well in such venues as the Journal of the Acoustical Society of America, or perhaps Hearing Research, or Music Perception.

Here are some more specific comments:

1. One of the conclusions from the various conditions is that listeners tend to prefer tone combinations that generate slow beating. This is an interesting observation, but as the authors seem to admit in the relevant paragraph of the discussion (top of p 30), there is no clear explanation for it. The speculation that it has somehow something to do with social bonding is particularly fanciful. I was also a bit puzzled about why in Fig 8 there is a preference for slow beats (generated by slightly mistuned tone in octave relationship) for complex tones but not for pure tones. If you take two pure tones of slightly differing frequencies, they will beat at the difference freq ($f_1 - f_2$). This is equivalent to a single tone with AM modulation at that difference rate. So why is there no preference for slow beats with pure tones?

2. Perhaps the most interesting finding is that when complex tones are generated with inharmonic partials, consonance ratings differ as shown in Fig 4. This effect implies that the timbre of tones (or at least the component of timbre that has to do with harmonicity/inharmonicity) can affect how combinations of them are perceived that cannot be predicted from the patterns seen with harmonic tones. This finding lends credence to older ideas about the relation between the timbres of certain instruments (such as Indonesian metallophones) and the scales that emerge from using those instruments. What I am missing in this whole discussion, though, is a consideration of how the timbre of the voice would influence consonance judgments. Even if you are a professional gamelan player, you will still have heard voices for most of your life, and so have all other hearing humans. So surely the timbre of the voice must have a huge influence; yet this is hardly discussed. There are passing references to the work of Bowling and Purves, but there is no discussion of the ideas that come from this group that scales (and not only Western ones) are derived from the acoustics of the human voice, which are the most salient periodic sounds in our environment.

3. Still with the idea that timbre can influence processing of scale relationships, there should have been a reference to Loui's recent paper on this topic (Cognition 2022)

4. Some of the results are a bit puzzling, and I found the discussion rather thin on how to interpret them. For instance, the data in Fig 9 show an interesting pattern for the triads, but I don't really understand how this pattern explains other phenomena, or even fits in with music cognition. Why do the authors think that the open chord (C-G-C) has a higher rating than the root-position major triad (C-E-G)? Are the inversions (C-F-A etc) equivalent to the root position? Are there some predictions that could come out of these relationships? It seems like a lot of work went into deriving these plots, so one would expect more to come out of it.

5. A small point that I found confusing is on p 30 where the authors write "While Western listeners might have certain tendencies towards liking or disliking certain acoustic attributes (e.g. roughness, slow beats), it is important to recognize that such appraisals are not necessarily universal. Musical enculturation seems to be important here, as evidenced for example by perceptual studies with the Tsimane' people, who seem (unlike Western listeners) to be indifferent towards harmonicity..." But in the study referred to, the Tsimane showed very similar dislike of roughness to the Americans. So I wasn't sure why the authors seem to be saying that roughness is just one more attribute, if it's still disliked even amongst those who don't care about harmonicity.

6. Finally I found it somewhat misleading in the abstract to say that >4000 participants were tested, first because as explained in the methods, there was no way to tell if the same person had done the test more than once, and second because that's the total number tested across all tasks, whereas on each task the number are a lot smaller.

Reviewer 1

The manuscript presents five large-scale online experiments investigating the influence of a sound's spectral content on subjective judgments of pleasantness. Pleasantness judgments are used as a proxy for the musical notion of consonance. The logic in the series of experiments is to test two classic models of consonance: an "interference" model, based on the theory of roughness introduced by Helmholtz, and the "harmonicity" model, with deeply established roots in music theory and more recent proposals by e.g. Terhardt. A "composite" model is also introduced, consisting of a simple linear combination of the predictions of the interference and harmonicity models. The experimental results show that the composite model is needed to capture effects of spectral content on pleasantness.

The question asked in the manuscript relates to perhaps one of the longest standing puzzles in (Western) music theory: what makes some combination of tones pleasant when sounded together, or con-sonant? As such, it will be of interest to the broad readership of the journal.

For full disclosure, I had already reviewed a previous version of the manuscript. In a nutshell, I think the way the study has been strengthened and enriched is very convincing. A strength of the manuscript is the sheer scale of the dataset: more than 4k participants overall. While the theoretical questions had already been framed in previous work, such as Sethares, 2005 and one of the authors' own review in 2020, the experimental data provided here are unique thanks to their scale and also thanks to some clever and original design choices. The composite model allows to make sense of the experimental results, providing a satisfactory resolution to seemingly contradictory findings favoring either interference or harmonicity. As an aside, I would also like to commend the authors for the very impressive Supplemental material they provide. In particular, the animated/audio version of the results would make for nice teaching materials or help non-musical readers to get the point of the study.

My only suggestions would revolve around how the composite model is introduced and specified, and whether it would be possible to get even more insight from it by considering inter-individual differences.

Response: We thank the reviewer for their favorable and thorough evaluation of our work! We address the remaining issues in what follows.

1) In terms of presentation, the models are all described in the Methods, with only a brief introduction in the main text. This makes for a more compact presentation of the experimental data, but it may also detract from the understanding of the insights provided by the modeling work. This is surely something to be left to the appreciation of the authors, but perhaps providing a bit more details on the models early on would be useful?

Response: We have now revised the Introduction to explain the models earlier. See the following passage:

We focus in particular on the two psychoacoustic models that performed best in a recent systematic evaluation of almost all extant consonance models (Harrison & Pearce, 2020). The first is the model of Hutchinson and Knopoff (1978), which supposes that dissonance derives from unpleasant interactions between neighboring partials in the frequency spectrum, potentially corresponding to the fast beats that occur when two tones of similar frequencies are superposed. Specifically, the model assigns a dissonance (or roughness) score for each pair of partials based on a parametric function that depends on the frequency difference between them, and then combines those scores additively. The second is the harmonicity model of Harrison and Pearce (2018), which supposes that chords become consonant when they align well with an idealized harmonic series. The harmonicity score is calculated by computing similarity scores between different idealized harmonic templates and a compact representation of the chord spectrum.

2) The parameters for the new models were fitted by hand, if I understood correctly (for instance, the improved way to take into account amplitude in Hutchinson & Knopoff, the way to introduce negative dissonance for slow beats, the relative weights of interference and harmonicity in the composite model). While the exercise has clearly been successful, I wonder if it would not be useful to optimize these parameters by performing a formal fit to the data.

Response: As suggested, we have now formally fitted these parameters to the data. To make the optimization process tractable and avoid local minima, we used the manually optimized parameters as a starting point and used a gradient-free optimizer to find improved versions of these parameters. As can be seen below, the parameters changed a little bit, but not much, and the qualitative patterns in the main paper remain unchanged. We have now updated the paper to use these numerically optimized parameters. We have added the following explanatory text to the Methods section:

Parameter optimization

Our new models include several key parameters, listed in Table 3. We set initial versions of these parameters based on a combination of theory and manual exploration of the data; we then numerically optimized these parameters using a gradient-free optimizer as described below.

Optimizing model parameters requires defining an objective function that is not too time-consuming to compute. We therefore excluded the triad experiments from the objective function, and solely modeled data from the dyad experiments.

For each dyad experiment, we computed consonance profiles for models and participants as described earlier. We found that correlation metrics were a poor measure of qualitative fit between models and participants, so we instead measured fit by comparing the peaks between the model and the participant profiles. Participant profile peaks were computed as previously; model profile peaks were obtained analogously, but instead of relying on bootstrapping to remove unreliable peaks (the model outputs are deterministic and hence can't be bootstrapped) we filtered excess peaks by increasing the peak-finding minimum depth parameter (β) from 0.01 to 0.05. We then defined model fit by looking at the overlap between participant peaks and model peaks. In particular, we characterized a pair of participant and model peaks as overlapping if they were separated by less than 2.67% of the overall interval range in that study (i.e. 0.4 semitones in the 15-semitone experiments, or 0.013 semitones in the 0.5-semitone experiments), and then computed the Jaccard similarity between the sets of participant and model peaks. The overall model fit was then calculated as the mean Jaccard similarity across all the dyad experiments.

We optimized this objective function as a function of the parameters listed in Table 3 using the 'subplex' algorithm (Rowan, 1990) as implemented in the NLOpt package (Johnson, 2023), and using the parameter bounds specified in Table 3. The model converged after 245 iterations to a mean Jaccard similarity of 0.467. The resulting model parameters (Table 3) are used for all visualizations and analyses involving the composite model.

Parameter	Value		Optimization bounds	
	Initial	Optimized	Lower	Upper
Harmonicity weight (relative to interference weight)	0.750	0.837	-1.000	1.000
Amplitude exponent in interference model	1.000	1.359	0.000	5.000
Slow-beat boundary (critical bandwidths)	0.100	0.096	0.000	1.000
Slow-beat pleasantness	1.500	1.632	0.000	5.000

Table 3. Model parameter optimization.

3) If point 2) were to be implemented, then this would perhaps open up a very interesting extension to the modeling work. It seems clear from Fig. 1B that there is a large individual variability in the data. The Methods convincingly show how meaningful average patterns were extracted from the raw judgements, and then most of the descriptions and discussions in the manuscript focus on the average trends (with the exception of the

musician/non-musician analyses). However, I wonder whether the authors would not be in an ideal position to also comment on how pleasantness judgments may systematically differ across participants. From the data, are there "clusters" of patterns of consonance? Could it be because listeners have different relative weights for interference and harmonicity in their very own composite model? I fully appreciate that the manuscript is already very rich, so I'd happily accept a comment to the effect of "this will be left for future studies" if this represents too much of an undertaking.

Response: We agree that this would be a very interesting approach. We investigated fitting models to individual participants, but found that the split-half reliability of the resulting models was very low due to the small number of trials per participant (median correlation: .39). This is attributable to our online experiment design, which had large numbers of participants but only a relatively small number of trials per participant. We therefore decided that we cannot do this approach justice at present; we are however excited to explore the topic in future work. We have now added a passage to the Discussion highlighting this possibility:

Consonance perception is known to vary between individuals, even when the individuals are drawn from a relatively homogeneous cultural group (McDermott et al., 2010; Popescu et al., 2019). It would have been interesting to analyze our own data at the participant level, but in practice our participant-level profiles had prohibitively low reliability due to the large stimulus space and the relatively small number of trials per participant (the median split-half reliability in Study 1 was .39). As a compromise, in the Supplementary Materials we break down the results by musicianship. We see that the results were generally similar across the participant groups, though a systematic investigation might well yield more interesting conclusions. An interesting future path is to collect more data per participant and then use unsupervised clustering methods to identify population subgroups with different response strategies (e.g., Pearce et al., 2010).

Reviewer 2

The paper is very clearly written and presents a novel and interesting experimental method, which uses finely grained microtonal tunings with harmonic and inharmonic spectra to decorrelate the potential influences of roughness and harmonicity on the pleasantness of musical dyads and triads. The findings are also very interesting – not only demonstrating the somewhat independent roles played by both mechanisms, but also suggesting some very interesting refinements to the classic roughness model; in particular, modifying the basic dissonance curve to account for the pleasantness of slow beats, which is a priori plausible and – for the first time – here experimentally demonstrated by using such finely grained pitch manipulations.

I strongly recommend this manuscript for publication due to the above-mentioned innovations in experiment method combined with interesting and novel findings related to human perception of harmony that these methodological innovations make possible, and a revised roughness model to account for these findings. The statistical and mathematical methods are well-conceived and rigorous. The methods and findings will certainly influence future investigations of the human experience of music.

Response: We are grateful for the reviewer's very positive and detailed evaluation! We address the remaining issues below.

I will now add a few minor suggestions. None of these need to be addressed, but the authors may find some of the comments useful.

p.2, l.60: "This would explain why scale systems across the world seem to have developed to favor harmonic pitch intervals" This is stated rather too definitively -- although common, it is not universally true (gamelan scales being a case in point)

Response: We revised this sentence as follows: "This would explain why many (though not all) scale systems across the world seem to have developed to favor harmonic pitch intervals"

p.3, l.89: "Previous consonance research has used stimuli drawn solely from discrete scales, in particular the Western 12-tone chromatic scale" This could be read as suggesting that all previous research has used only 12-TET, which of course is not true.

Response: We revised this sentence as follows: "Previous consonance research has used stimuli drawn solely from discrete scales, most commonly the Western 12-tone chromatic scale"

p.5, l.160: "Other possible synonyms exist (e.g., smoothness, purity, harmoniousness), but in practice human ratings tend to be highly correlated across these synonyms (Lahdelma & Eerola, 2020)" I wonder if this is slightly overstating things. There are important differences noted in the cited paper between some terms; in particular, how "tension" and "pleasantness" are, respectively, stable or change across differing musical familiarities and expertise. "Tension" is not mentioned in the parentheses nor other commonly used synonyms such as "stability".

Response: Thanks for pointing this out. As suggested we have added tension and stability to the list, and have replaced 'highly correlated' with 'correlated': "Other possible synonyms exist (e.g., smoothness, purity, harmoniousness, tension, stability), but in practice human ratings tend to be correlated across these synonyms (Lahdelma & Eerola, 2020)".

p.26, l.669: "They also provide an empirical justification for certain practices in the experimental music tradition of 'Dynamic Tonality' (Plamondon et al., 2009)" I wonder if

Sethares, W. A., Milne, A. J., Tiedje, S., Precht, A., and Plamondon, J. (2009). Spectral tools for Dynamic Tonality and audio morphing. *Computer Music Journal*, 33(2):71–84 would be a better or additional citation here?

Response: Thanks for suggesting this, we have now added this citation to the text.

p.29, l.726: "The preferences we document for slight inharmonicity moreover shed new light on the historical development of Western tuning systems: they indicate that the slight impurities of systems such as mean-tone and equal temperament may not detract from subjective pleasantness, as is typically assumed, but instead positively contribute to it." I think this is an interesting point and certainly applicable to several meantone tunings. But for 12-TET around middle-C, the beating of the lower tone's 5th harmonic and the upper tone's 4th will be about 10Hz and so somewhat above the range of pleasant slow beating. Historically speaking, meantone probably gradually adapted into 12-TET so these pleasant impurities at the start may have eased that path even though the end-point is not really consistent with the slow-beating hypothesis.

Response: We agree with the reviewer that it is good to be more specific here. In particular, we like the idea of sharing representative beating frequencies for 12-TET as well as meantone. We're not sure what pitch interval the reviewer was referring to, but we think it makes sense to study a perfect fifth on middle C, i.e. C4-G4, in which case we get a beating frequency of 2.43 Hz for quarter-comma meantone and 0.889 Hz for 12-TET, which make more sense than 10 Hz for the slow-beats hypothesis. We have included the logic below in case the reviewer wants to check our working.

- A perfect fifth in quarter-comma meantone corresponds to ~ 696.578 cents.
 - If C4 is 261.6256 Hz, then G4 is 391.2214 Hz.
 - The 3rd harmonic of C4 is 784.8767 Hz.
 - The 2nd harmonic of G4 is 782.4427 Hz.
 - The difference tone is then 2.433949 Hz.
- A perfect fifth in 12-TET corresponds to 700 cents.
 - If C4 is 261.6256 Hz, then G4 is 391.9954 Hz.
 - The 3rd harmonic of C4 is 784.8767 Hz.
 - The 2nd harmonic of G4 is 783.9909 Hz.
 - The difference tone is then 0.8858235 Hz.

We have updated the corresponding part of the paper:

The preferences we document for slight inharmonicity moreover shed new light on the historical development of Western tuning systems: they indicate that the slight impurities of systems such as mean-tone and equal temperament may not detract from subjective pleasantness, as is typically assumed, but instead positively contribute to it by creating pleasant slow beating. For example, the perfect fifth C4-G4 elicits no perceptible beats when played in just intonation, but elicits

pleasant slow beats at 2.43 Hz if played in quarter-comma meantone tuning, and at 0.89 Hz if played in 12-tone equal temperament.

p.36, l.949: It would be more conventional for sin to be in an upright font; e.g., by using `|sin` in LaTeX

Response: Fixed.

p.38, l.1001: "How well does the sound match the following word (pay attention to subtle differences): pleasant" What motivates this specific choice of wording? It suggests that there are likely to be other words than "pleasant" for the comparisons. Were there? If so they should be reported here even if not discussed further. If not, is it problematic that the participants may have been waiting for a different question to appear (which never did)? Is there an advantage to asking the question in this way rather than just providing a rating scale with something like "very unpleasant" and "very pleasant" at the two ends? It would be useful to provide some reasoning behind the wording question. (Overall this is a very minor comment, but I was struck by the unusual wording of the question.)

Response: This specific wording was motivated by the Gibbs Sampling with People paradigm which was conceived as a tool for exploring different adjectives (Harrison et al., NeurIPS, 2020). A previous version of this paper included GSP results for different adjectives, but we eventually decided to focus on reporting only the adjective of ‘pleasantness’ for the sake of brevity, and leave these other results for future work. We now clarify this point in the paper:

We note that while we did not collect ratings for adjectives other than “pleasant” in this study, the idea here was to use a wording that would allow the dense rating paradigm to be consistently applied across multiple adjectives in future work (as in Harrison et al., 2020).

p.47, l.1245: "We smoothed all model profiles using the same process as for the behavioral profiles" This makes no sense to me. Why were the model's predictions smoothed? The purpose of smoothing is to reduce "high-frequency" noise so as to reveal an underlying distribution. But there is no stochastic element to the predictions of roughness or harmonicity made by any of these models, so I cannot see any justification for smoothing their results. Can this be explained? Or, if there is no explanation, perhaps the smoothing should not be applied?

Response: The primary motivation for smoothing the behavioral data is indeed to reduce high-frequency noise, but a consequence is that each point of the resulting behavioral profile estimates the local average of the pleasantness of that neighborhood, rather than pleasantness of that precise point. If we want to predict this behavioral profile using a model, then the best strategy is to follow the same approach: predict the pleasantness of the neighborhood and average it, or in other words, smooth the model profile

analogously to the behavioral profile. If we don't do this, then small-scale fluctuations in the model output (e.g. due to slow beating) will look inconsistent with the behavioral profile, even though the behavioral profile has no chance to display such fluctuations because of the smoothing applied.

This explanation felt too wordy for the paper itself so we have instead updated the highlighted passage in the text as follows: "We smoothed all model profiles using the same process as for the behavioral profiles to maximize comparability between the two sets of profiles."

p.55, l.1462: "The two models are combined additively with a weight of -1 for the interference model and +0.75 for the harmonicity model (weights chosen on the basis of manual exploration)." Excellent to see a single model (identical parameter values) applied to every data set (particularly given that this has been abused resulting in gross overfitting in some music cognition research, which will remain nameless for the purpose of this review!). However, I was surprised to see the weights had been set by "manual exploration", which I presume means changing the relative weights until the curves seem – by eye – to fit better. Given the statistical and mathematical rigour of most of the paper, couldn't there be a more systematic way to optimise this relative weight? Or is there a good argument for not employing a more systematic approach?

Response: We agree. We have now implemented a numeric optimization procedure as suggested (see response to Reviewer 1).

p.55, l.1467: "Note that we plot the final version of the composite model throughout the paper, rather than plotting incremental versions as motivated by each experiment, to ensure that later model changes do not spoil predictions for earlier experiments." Something similar but less complete is mentioned earlier in the paper. However, it wasn't until I read this part that I was sure whether the weights of the two components had been kept the same or just the adjustments related to using amplitude (instead of intensity) and the modification of the dissonance curve. I would suggest making it clear when first mentioned that the weights are also kept the same because this is an important positive aspect of this article's analyses.

Response: Thank you for pointing this out, we now explicitly state that in the text: "Additionally, we plot results from a new 'composite' model that comprises a simple weighted average of updated versions of the interference and harmonicity models, with weights fixed throughout the paper, [...]".

Reviewer 3

In this paper the authors perform a series of psychoacoustical online experiments in which various combinations of complex and pure tones are presented in order to explore the concept of dissonance and its dependence on various factors, including harmonicity of the constituent tones, spectral shape (roll-off), and presence/absence of certain harmonics. The methodology has some advantages (sample size, mostly, but also ability to sample across a wide parameter space); but it also has some weaknesses (e.g. mixing of between vs within-subject variance, which makes it hard to identify individual difference; as well as poor control over attention, sound level, hearing status, and other details that are normally highly controlled in traditional psychophysics). But for the purpose of providing data over wide ranges of parameter combinations, as done here, it seems to work remarkably well. The results demonstrate some interesting interactions, especially with regards to how harmonicity (stretching vs compressing the harmonics) influences judgments of consonance. Another important aspect of the study is that two models (harmonicity vs interference) are compared explicitly, which has rarely (or maybe never) been done in a formal way; plus they also develop their own hybrid model combining some features of each, which turns out to fit the data better than either of the ones most commonly espoused in the literature.

Despite the technical tour-de-force, and the paper's many good features, I miss the bigger impact that these findings might have. It is certainly of interest to a small segment of the scientific community (among whom I would count myself) interested in the psychoacoustics of consonance. But I failed to see the conceptual advance that would have implications for other domains of broader interest, including speech, or even music perception. In the case of music, the role of consonance/pleasantness in music is hugely modulated by context, but the approach taken here eschews all such considerations, which is fine for purposes of identifying the basic relations but not very useful for inferring how these relations actually play out in real music. I can see this paper fitting very well in such venues as the Journal of the Acoustical Society of America, or perhaps Hearing Research, or Music Perception.

Response: We acknowledge that we could have made the paper's impact clearer. The original manuscript focused on arguing how (a) our consonance studies help explain scale evolution across the world, and how (b) our methods have many potential applications in domains such as psychology, psychophysics, and auditory perception. We do believe that both (a) and (b) are substantial contributions, but we have now added two paragraphs to the Discussion which we believe further improve our case.

First we add a paragraph that acknowledges how context is an important modulator of consonance perception, but emphasizes how the consonance of individual chords is nonetheless essential for explaining a remarkable range of musical structures:

We studied the consonance of isolated chords, but in many musical styles consonance is treated as a dynamic phenomenon; for example a dissonant note may be ‘prepared’ by a previous chord and ‘resolved’ by subsequent melodic movement. These temporal aspects of consonance are very interesting in their own right and worthy of further research (e.g. Johnson-Laird et al., 2012; Wright & Bregman, 1987). Nonetheless, we have shown how experiments with isolated chords are very effective for elucidating the fundamental relationships between intervallic structure and consonance. These relationships have shaped the treatment of musical pitch in a vast number of musical styles across the world, guiding the evolution of scales, harmonic vocabulary, and harmonic progressions (Chiba et al., 2019; Gill & Purves, 2009; Hall, 1973; Huron, 1994, 2001).

We then add a paragraph explaining how consonance links to the processing of non-musical sounds and hence to auditory perception at large:

While we have focused on identifying acoustic features that drive pleasantness in musical contexts, it is worth considering how these features might occur in other auditory contexts, and what the implications might be for the subjective appraisal of non-musical sounds. Harmonicity is a characteristic feature of vocalizations, and it has been suggested that social animals should find such sounds attractive because they (often) indicate the presence of conspecifics (Bowling & Purves, 2015; Bowling et al., 2018). Roughness is meanwhile a feature of vocalizations indicating distress, in particular screams; it is evident that such sounds should cause discomfort because they are indicative of nearby danger (e.g. Arnal et al., 2015). It is less clear why slow beats might sound attractive, but one explanation is that these fluctuations are suggestive of relaxed (i.e. non-distressed) vocalizations, and hence imply a safe environment. These hypotheses would imply that consonance is not simply a musical phenomenon, but instead is the (culturally shaped) musical manifestation of deep principles from general auditory perception.

Together with our pre-existing detailed arguments about scale evolution (see paragraph 2 of the Discussion) and non-musical applications of our methods (see last paragraph of the Discussion), we believe that our paper now articulates a significant and broad impact.

Here are some more specific comments:

1. One of the conclusions from the various conditions is that listeners tend to prefer tone combinations that generate slow beating. This is an interesting observation, but as the authors seem to admit in the relevant paragraph of the discussion (top of p 30), there is no clear explanation for it. The speculation that it has somehow something to do with social bonding is particularly fanciful.

Response: We have now removed this social bonding speculation and left the other (less fanciful) hypothesis.

I was also a bit puzzled about why in Fig 8 there is a preference for slow beats (generated by slightly mistuned tone in octave relationship) for complex tones but not for pure tones. If you take two pure tones of slightly differing frequencies, they will beat at the difference freq ($f_1 - f_2$). This is equivalent to a single tone with AM modulation at that difference rate. So why is there no preference for slow beats with pure tones?

Response: We think the reviewer may have confused the octave for the unison. Indeed, pure tones played at a slightly detuned unison (e.g. 400 Hz, 403 Hz) will produce slow beating. However, pure tones played at a slightly detuned octave (e.g. 400 Hz, 803 Hz) would not induce the same slow beats.

2. Perhaps the most interesting finding is that when complex tones are generated with inharmonic partials, consonance ratings differ as shown in Fig 4. This effect implies that the timbre of tones (or at least the component of timbre that has to do with harmonicity/inharmonicity) can affect how combinations of them are perceived that cannot be predicted from the patterns seen with harmonic tones. This finding lends credence to older ideas about the relation between the timbres of certain instruments (such as Indonesian metallophones) and the scales that emerge from using those instruments. What I am missing in this whole discussion, though, is a consideration of how the timbre of the voice would influence consonance judgments. Even if you are a professional gamelan player, you will still have heard voices for most of your life, and so have all other hearing humans. So surely the timbre of the voice must have a huge influence; yet this is hardly discussed.

Response: Our experiment with the bonang timbre (Study 2C; Figure 5) actually takes that into account. Note that the way the dyads were constructed was by taking one of the tones to be a harmonic complex tone, and the other tone to be of the synthetic bonang timbre, precisely to emulate the interaction between the human voice and the instrument. We now explicitly highlight this in the text: “In particular, he proposed that the inharmonic slendro scale might be explained in terms of the consonance profile produced by combining a harmonic complex tone with a bonang tone, as in our own experiment, emulating the interaction between the human voice and the bonang.”, and in Table 1.

It is also worth noting that Figure 5 includes predictions from a harmonicity model, capturing a general preference for harmonic sounds that may in part derive from familiarity with human vocalizations. This forms part of the resulting composite consonance model.

There are passing references to the work of Bowling and Purves, but there is no discussion of the ideas that come from this group that scales (and not only Western ones)

are derived from the acoustics of the human voice, which are the most salient periodic sounds in our environment.

Response: This hypothesis was originally mentioned in the Discussion but we have now extended the treatment of this passage (see above).

3. Still with the idea that timbre can influence processing of scale relationships, there should have been a reference to Loui's recent paper on this topic (Cognition 2022)

Response: Thanks for the suggestion, we have added the reference to the introduction as we agree that it is highly relevant: “Furthermore, recent decades of psychological studies seem to show that timbral manipulations do not qualitatively affect consonance judgments (Friedman et al., 2021; McDermott et al., 2010; McLachlan et al., 2013; Nordmark & Fahlén, 1988; Parncutt et al., 2023; though see Loui, 2022, for a study of the effect of timbre on the statistical learning of melodic grammars).”

4. Some of the results are a bit puzzling, and I found the discussion rather thin on how to interpret them. For instance, the data in Fig 9 show an interesting pattern for the triads, but I don't really understand how this pattern explains other phenomena, or even fits in with music cognition. Why do the authors think that the open chord (C-G-C) has a higher rating than the root-position major triad (C-E-G)? Are the inversions (C-F-A etc) equivalent to the root position? Are there some predictions that could come out of these relationships? It seems like a lot of work went into deriving these plots, so one would expect more to come out of it.

Response: The observation that the rating for the open chord (C-G-C) is higher than that of the major triad (C-E-G) is predicted by both interference and harmonicity theories (Figure 10). This is consistent with Western music theory: the former only comprises ‘perfect’ consonances (fourths and fifths), whereas the latter also includes ‘imperfect’ consonances (i.e. thirds).

We now added to the text the following clarification and interpretation: “A further consonance hotspot corresponds to the 'bare fifth', or 'power chord', a chord comprising the fifth but no third. This chord is common in both medieval music and rock music.”

More broadly, we emphasize that the purpose of Study 5 was to test how our findings regarding stretched and compressed timbre generalize to triads. This constitutes an important conceptual replication of our dyadic findings; it also provides a good validation of our GSP technique, which is necessary for dealing with this more complex stimulus space.

To clarify the contribution of the triads experiment we added this sentence in the discussion:

The triad space is much more complex than the dyad space, so it serves as a “hard” test for the composite model's predictions.

5. A small point that I found confusing is on p 30 where the authors write "While Western listeners might have certain tendencies towards liking or disliking certain acoustic attributes (e.g. roughness, slow beats), it is important to recognize that such appraisals are not necessarily universal. Musical enculturation seems to be important here, as evidenced for example by perceptual studies with the Tsimane' people, who seem (unlike Western listeners) to be indifferent towards harmonicity..." But in the study referred to, the Tsimane showed very similar dislike of roughness to the Americans. So I wasn't sure why the authors seem to be saying that roughness is just one more attribute, if it's still disliked even amongst those who don't care about harmonicity.

Response: We see the confusion and have now replaced the example attributes (“roughness, slow beats”) with “harmonicity”, which is indeed an example of a culturally variable attribute.

"While Western listeners might have certain tendencies towards liking or disliking certain acoustic attributes (e.g. harmonicity)"

6. Finally I found it somewhat misleading in the abstract to say that >4000 participants were tested, first because as explained in the methods, there was no way to tell if the same person had done the test more than once, and second because that's the total number tested across all tasks, whereas on each task the number are a lot smaller.

Response: Participants were not allowed to participate in the same experiment more than once. They could, however, re-participate across experiments. To address the reviewer's concern we now revised the abstract to report the number of unique human judgments (235,440), thereby avoiding the ambiguity.

Reviewers' Comments:

Reviewer #1:

Remarks to the Author:

The authors have satisfactorily addressed all of the issues that had been raised in my previous review. Even though the individual fitting of models to participants has proved to be technically unfeasible, I am happy to hear that this will be considered for future work. In any case, congratulations for this excellent contribution.

I have also been asked to comment on the points raised by Reviewer 3. Their general comment was about the suitability of the manuscript to a high-profile journal, rather than a specialty journal. Although this is ultimately an editorial decision, my opinion is that the sheer size of the dataset combined with the new unifying model, which has exciting potential for future developments, make the contribution appealing to a broad audience. In the Rebuttal, the authors also point to their methodological contribution, and to addition they made to the Discussion arguing for the relevance of their findings to comparative theories of music as well as appraisal of non-musical sounds.

The specific comments of Reviewer 3 have all been addressed in details.

Comment 1) asked about why slow beats are pleasant? As requested, the authors have removed the speculative interpretation based on social bonding. They introduced a new idea about slow beats signaling "relaxed" vocalizations, which I am not sure I find necessarily more convincing: wouldn't the most relaxed vocalizations contain no beats at all? Thinking a bit more about the issue, I could come up with other ideas. For instance, it may be that being used to the equal-tempered scale contributes to appreciating slow beats, as many equal-tempered intervals will produce such beats. Even though preference did not match equal temperament in Fig 8, this could still contribute to a composite model. Also, if I'm not mistaken, piano tuners deliberately introduce slow beats between strings of the same note to make the sound "more interesting" - perhaps they know something. But in any case, all of this is speculation and I believe the authors are entitled to present their own pet theory, if it is clearly marked as such.

Comments 2) and 3) were discussion points about the relation of timbre and consonance. Useful additions have been made following the recommendations.

Comment 4) related to the interpretation of the triad experiment. The results do generally match the model predictions, and the specific musical examples brought forward by Reviewer 3 are mentioned in the revised manuscript.

Comment 5) has been dealt with by avoiding to mention Tsimane's results about roughness.

The issue of unique participants vs trials raised in comment 6) has been clarified.

Reviewer #2:

Remarks to the Author:

I am glad to see that a numerical optimisation has been used for the two model weights. This is good enhancement.

There seems to be some confusion about the point I was making regarding beating in 12-TET, meantone, and Pythagorean tuning systems. I was referring to beating between the 5th partial of the lower tone and the 4th partial of the higher tone in a major third. Why the major third? Because the motivation for the meantone system was to substantially improve the thirds (with only a small detuning of the fifths). By definition, the beating between the respective 5th and 4th partials of a

major third in 1/4-comma meantone is zero. In a meantone tuning with slightly larger fifths (e.g. 1/5-comma or 1/6-comma meantone), the beating of the 5th and 4th partials in a major third will be in the pleasant range. But when the fifths get as large as they are in 12-TET, the beating is actually quite fast -- for C4-E4, those partials beat at 10.38Hz!

So the discussion needs to be more nuanced here. The facts are that: with Pythagorean tuning, you have fifths (partials 2 and 3) not beating, but very rapid beating with thirds (partials 4 and 5). With 1/4-comma meantone, you have thirds with no beating, and fifths with slow beating. With meantone tunings with fifths slightly larger than 1/4-comma, you have both thirds and fifths with slow beating. With 12-TET, you have slow-beating fifths, but the thirds are now verging into (fully into?) unpleasant fast beating. It is possible that the meantone tunings forged a path towards 12-TET, which ultimately "wins" historically due to its structural simplicity. However, I realise it may be difficult to explain the above in a way that is not excessively complex and log-winded (particularly given that it is really just a conjecture). But I do feel the revised version, which focusses only on fifths doesn't really make sense from a historical perspective. So there should be some revision of this text.

Everything else looks good.

Point-by-point response to reviewers

Reviewer #1 (Remarks to the Author):

***1.1** ‘The authors have satisfactorily addressed all of the issues that had been raised in my previous review. Even though the individual fitting of models to participants has proved to be technically unfeasible, I am happy to hear that this will be considered for future work. In any case, congratulations for this excellent contribution.’*

1.1 Response: Thank you!

***1.2** ‘I have also been asked to comment on the points raised by Reviewer 3. Their general comment was about the suitability of the manuscript to a high-profile journal, rather than a specialty journal. Although this is ultimately an editorial decision, my opinion is that the sheer size of the dataset combined with the new unifying model, which has exciting potential for future developments, make the contribution appealing to a broad audience. In the Rebuttal, the authors also point to their methodological contribution, and to addition they made to the Discussion arguing for the relevance of their findings to comparative theories of music as well as appraisal of non-musical sounds.’*

‘The specific comments of Reviewer 3 have all been addressed in details.’

‘Comment 1) asked about why slow beats are pleasant? As requested, the authors have removed the speculative interpretation based on social bonding. They introduced a new idea about slow beats signaling “relaxed” vocalizations, which I am not sure I find necessarily more convincing: wouldn’t the most relaxed vocalizations contain no beats at all? Thinking a bit more about the issue, I could come up with other ideas. For instance, it may be that being used to the equal-tempered scale contributes to appreciating slow beats, as many equal-tempered intervals will produce such beats. Even though preference did not match equal temperament in Fig 8, this could still contribute to a composite model. Also, if I’m not mistaken, piano tuners deliberately introduce slow beats between strings of the same note to make the sound “more interesting” - perhaps they know something. But in any case, all of this is speculation and I believe the authors are entitled to present their own pet theory, if it is clearly marked as such.’

1.2 Response: We revised the relevant paragraph in the Discussion to highlight the speculative nature: “It is less clear why slow beats might sound attractive, but one speculation is that these fluctuations are suggestive of relaxed (i.e. non-distressed) vocalizations, and hence imply a safe environment. Alternatively, slow beating might be preferred simply because it makes the sounds more interesting. These hypotheses would imply that consonance is not simply a musical phenomenon, but instead is the (culturally shaped) musical manifestation of deep principles from general auditory perception.

However, these preferences are likely also to be shaped by musical experience; for example, familiarity with Western music may well encourage disliking of roughness and liking of slow beats.”

1.3 *‘Comments 2) and 3) were discussion points about the relation of timbre and consonance. Useful additions have been made following the recommendations.*

‘Comment 4) related to the interpretation of the triad experiment. The results do generally match the model predictions, and the specific musical examples brought forward by Reviewer 3 are mentioned in the revised manuscript.

‘Comment 5) has been dealt with by avoiding to mention Tsimane’s results about roughness.

‘The issue of unique participants vs trials raised in comment 6) has been clarified.’

1.3 Response: We thank the Reviewer for their careful evaluation of our response to Reviewer 3.

Reviewer #2 (Remarks to the Author):

2.1 *‘I am glad to see that a numerical optimisation has been used for the two model weights. This is good enhancement.’*

2.1 Response: Thank you for suggesting this.

2.2 *‘There seems to be some confusion about the point I was making regarding beating in 12-TET, meantone, and Pythagorean tuning systems. I was referring to beating between the 5th partial of the lower tone and the 4th partial of the higher tone in a major third. Why the major third? Because the motivation for the meantone system was to substantially improve the thirds (with only a small detuning of the fifths). By definition, the beating between the respective 5th and 4th partials of a major third in 1/4-comma meantone is zero. In a meantone tuning with slightly larger fifths (e.g. 1/5-comma or 1/6-comma meantone), the beating of the 5th and 4th partials in a major third will be in the pleasant range. But when the fifths get as large as they are in 12-TET, the beating is actually quite fast -- for C4-E4, those partials beat at 10.38Hz!*

‘So the discussion needs to be more nuanced here. The facts are that: with Pythagorean tuning, you have fifths (partials 2 and 3) not beating, but very rapid beating with thirds (partials 4 and 5). With 1/4-comma meantone, you have thirds with no beating, and fifths with slow beating. With meantone tunings with fifths slightly larger than 1/4-comma, you have both thirds and fifths with slow beating. With 12-TET, you have slow-beating fifths, but the thirds are now verging into (fully into?) unpleasant fast beating. It is possible that the meantone tunings forged a path towards 12-TET, which ultimately "wins" historically due to its structural simplicity. However, I realise it may be difficult to

explain the above in a way that is not excessively complex and log-winded (particularly given that it is really just a conjecture). But I do feel the revised version, which focusses only on fifths doesn't really make sense from a historical perspective. So there should be some revision of this text.'

2.2 Response: We agree. We revised the Discussion accordingly: “The preferences we document for slight inharmonicity moreover have implications for the historical development of Western tuning systems: they indicate that the slight impurities of systems such as mean-tone and equal temperament may not always detract from subjective pleasantness, as is typically assumed, but can positively contribute to it by creating pleasant slow beating. For example, the perfect fifth C4-G4 elicits no perceptible beats when played in just intonation, but elicits pleasant slow (2.43 Hz) beats between the 2nd and 3rd harmonics if played in quarter-comma meantone tuning, and at 0.89 Hz if played in 12-tone equal temperament. In contrast, the major third elicits no perceptible beats in either just intonation or quarter-comma meantone tuning, but elicits fast (10.38 Hz) beats between the 4th and 5th harmonics in 12-tone equal temperament, a speed at which the beating will likely start to feel unpleasant again.”

2.3 *'Everything else looks good.'*

2.3 Response: Thank you!